



# Cloud history changes water-ice-surface interactions of oxide mineral aerosols (e.g. Silica)

Ahmed Abdelmonem[1*], Sanduni Ratnayake[2], Jonathan D. Toner[3] and Johannes Lützenkirchen[2]

[1]Institute of Meteorology and Climate Research - Atmospheric Aerosol Research (IMKAAF), Karlsruhe Institute of Technology (KIT), 76344 Eggenstein-Leopoldshafen, Germany
[2]Institute of Nuclear Waste Disposal (INE), Karlsruhe Institute of Technology (KIT), 76344 Eggenstein-Leopoldshafen, Germany
[3]Department of Earth & Space Sciences, University of Washington, Seattle, WA 98195, USA
* Correspondence to: A. Abdelmonem (ahmed.abdelmonem@kit.edu)

## Abstract

Mineral aerosol particles can act as ice nucleators, and many insights have been obtained on water freezing as a function of mineral surface properties such as the charge or morphology. Previous studies have mainly focused on pristine samples, despite the fact that under natural atmospheric conditions, aerosol particles age. For example, an aerosol-containing cloud droplet can go through different freeze-melt cycles, so that not only the aerosol surface structure may change, but also the ionic strength and pH of the cloud droplet. The potential variation of the surface properties of an ice nucleating particle during its residence in the atmosphere has been largely overlooked. Here, we use an environmental cell in conjunction with nonlinear spectroscopy (second-harmonic generation) to study the effect of freeze-melt processes on aqueous chemistry at silica surface at low pH. We found that the successive freeze-melt cycles disrupt the dissolution equilibrium, substantially changing the surface properties, giving rise to marked variations in the interfacial water structure and the ice nucleation ability of the surface. The degree-of-order of water molecules, next to the surface at a specific temperature, decreases and then increases again with sample aging. The water ordering–cooling dependence and ice nucleation ability improve continuously.



# 1 Introduction

Water- and ice-mineral interactions play vital roles in the atmosphere as well as in food, pharmaceutical, construction, chemical and other industries. The chemical and morphological properties of aerosol surfaces play direct and indirect roles in the climate system. For atmospheric questions, these systems should be looked up under non-equilibrium conditions, as in

nature, atmospheric constituents are not in rest. Understanding the role of surface properties and their potential variations under atmospheric conditions remains challenging. For example, the ice nucleation (IN) ability of an ice nucleating particle (INP) may be influenced by the change in surface properties due to aging processes in ice nuclei (Coluzza et al., 2017) or other secondary ice processes (Zipori et al., 2018). Solutes are able to affect freezing (Zobrist et al., 2008) and as recently shown even in minor concentration (Whale et al., 2018). Water molecules may heterogeneously crystallize next to an INP

surface forming one of the various ice polymorphs with different physical properties depending on the atmospheric conditions (e.g. degree of supersaturation) and/or surface properties (Parambil et al., 2014). This in turn affects their interaction with radiation in the atmosphere for instance and has a well-known impact on the energy budget of the planet (Steiner et al., 2013). In mixed phase clouds, for example, different polymorphs of ice can scatter light at different angles and hence affect the radiation balance. Clouds and cloud formation are significant perturbations in the climate models. This

means that understanding the elemental processes of IN and the role of surface properties have become a demand to counteract the climate change, for example by controlling the IN processes in atmosphere.

IN and crystallization processes are highly dependent on the initial stages where few molecules or ions start to form a tiny crystalline nucleus in the liquid (Sosso et al., 2016; Cox et al., 2007; Cox et al., 2013). It is not possible to predict the time and place this nucleus will form in a real system. This complicates interpretations of experimental nucleation studies.

Additionally, a pre-adsorption of small impurities on the aerosol surface can dramatically affect the IN process. On the other hand, theoretical and simulation studies are complex not only because of the additional complication of describing the surface and the surface-solute interactions, but also because of potential surface modifications under atmospheric conditions. Nucleation and crystallization are very active areas of current research.

Most IN processes occurring in the atmosphere are, in fact, heterogeneous, due to the omnipresence INPs. Hoose and Möhler

have reported the wide scatter in IN efficiencies of same aerosol particles collected from different sources and examined in different laboratories (Hoose and Mohler, 2012). There have been previous efforts to understand the influence of surface properties on water freezing such as the effects of charge, e.g., on ice formation by aluminum oxide (Anim-Danso et al., 2016; Abdelmonem et al., 2017) or morphology, e.g., for Feldspar (Kiselev et al., 2017) and Kaolinite (Wang et al., 2016). One common effect that appears to have received relatively little attention or is identified as a topic for future studies

(Coluzza et al., 2017)  is the potential variation of surface properties of a given INP over its residence time in the atmosphere. An aerosol particle in a cloud droplet is to some extent very similar to a particle in a solution. The ionic strength and pH of this solution may change due to different reasons. For example, an insoluble aerosol particle like silica may adsorb soluble salts like NaCl or acids like HCl or $H_2SO_4$ from the atmosphere or even at the surface of the earth before being aerosolized. Under suitable supersaturation conditions, water molecules in the atmosphere condense on the surface of that

aerosol particle forming a so-called cloud droplet. Actually it should rather be called a solution droplet. The ionic strength as well as the acid/basic character of this solution droplet is determined by the concentrations of all involved salts, acids and bases, respectively. These may affect the interfacial chemistry at the solid surface, e.g. through dissolution (Lis et al., 2014). According to the literature, dissolution of silica may change the ionic strength of interfacial water which may in turn change the surface charge and also screen this charge by nearby ions (Seidel et al., 1997; Schaefer et al., 2017). Silica dissolution

under non-equilibrium conditions, namely flow, has been recently discussed in some details (Schaefer et al., 2018). They demonstrated that silica equilibrates at the interface with pure water at around 1mM of ionic strength. In the atmosphere, the rate of surface properties changes, however, is variable depending on the change in the droplet size upon further condensation or partial evaporation. Partial evaporation may drive the system to rather extreme changes of mineral INP



surface properties. A typical mixed phase droplet of a size of 50 μm with a pH value of 8 (assuming an initial 1 μM NaOH solution) may evolve to pH = 10 when the size is reduced to its half (i.e. 25 μm). At this pH value, silica for example undergoes significant dissolution with fast rates (Hiemstra and van Riemsdijk, 1990). A possible worst case scenario could be near complete evaporation under dry conditions which will drive pH and ionic strength to extreme values. Besides

dissolution this can cause changes in morphology, both of which are known to affect the ordering of water molecules and ice nucleation ability. In this manuscript, we intend to demonstrate such effects on the structure of water that arise from aging of the surface and the variation of the solution composition due to freezing/melting. We investigate the possible variation of the water- and ice- aerosol surface interactions with only one parameter in its environment which is the pH of the cloud droplet. We use fused silica surface as a model of mineral oxide aerosols.

Silicon is the most common element on earth after oxygen. In principle it may be present in the form of crystalline or amorphous $SiO_2$ or Si containing minerals. Both crystalline and amorphous forms of $SiO_2$ have been widely studied (Iler, 1979; Bergna and Roberts, 2005) also with interfacial behavior including silica/water interface under static conditions using non-linear spectroscopy techniques (Ong et al., 1992; Ostroverkhov et al., 2004, 2005; Jena and Hore, 2009; Jena et al., 2011; Azam et al., 2012, 2013; Ohno et al., 2016; Dalstein et al., 2017; Darlington et al., 2017; DeWalt-Kerian et al., 2017;

Schaefer et al., 2017; Boamah et al., 2018; Schaefer et al., 2018), potentiometric titration (Karlsson et al., 2001; Dove and Craven, 2005), Atomic Force Microscopy(Morag et al., 2013), or X-ray Photoelectron Spectroscopy(Brown et al., 2016). However, studies under non-equilibrium conditions are rare (Lis et al., 2014; Gibbs-Davis et al., 2008; Schaefer et al., 2018). Here, we are interested in fused silica, which is an amorphous form of $SiO_2$, under non-equilibrium conditions. From the point of view of surface chemistry silica has often been considered unusual on the one hand side because its surface charge

characteristics are different from those of many other mineral oxides in the sense that classical silica charging curves involve a plateau at zero charge below about pH 5.5. On the other hand silica exhibits unusual aggregation behavior and would remain stable under conditions, where other mineral oxides would aggregate. A hairy like structure at the surface of colloidal silica has been advocated to explain this (steric stabilization). In the context of surface charging, silica has been often associated with a gel-like layer at its surface. The latter two aspects have been recently addressed in some detail (Schrader et

al., 2018). It was concluded from surface force measurements that neither indications for the hairy layer that has often been advocated as the explanation for the unusual stability nor evidence for the formation of a gel layer could be observed. Instead the unusual behavior was explained by the heterogeneity of the surface which in turn arises from the presence of silanol (hydrophilic) and siloxane (hydrophobic) surface groups. To what extent these issues are varying with chemical conditions (i.e. exposure of a given surface for long times to a given solution, which might be more or less harsh, or exposure of a given

surface to changing conditions over a long time) has not been frequently studied, in particular when changing temperatures between -40 and 25 °C are concerned. In this context and taking up on the discussion above, the freezing and melting processes involving a dilute solution at room temperature will result in a solution during freezing and melting in contact with the surface with a very high concentration of salt or acid/base (see SI and Fig. S1).

The rates of quartz dissolution at 70 °C at pH 0 are as fast as at pH 8 to 10 (depending on the salt level) according to a

comprehensive model by (Hiemstra and van Riemsdijk, 1990). The trend to increasing rates at very low pH (the rate minimum is at pH 3) is partially visible in some of the experimental data, and extrapolation suggests very fast dissolution at pH -1. The rates are also affected by the temperature, so that we here only use the available data and analogies to illustrate that dissolution is fast at very low pH values. This causes changes to the surface and should find repercussions in the surface properties. Surfaces of particles in the atmosphere undergo this kind of freeze/melt cycles as well and therefore aged surfaces

are probably much more relevant to IN than the freshly prepared samples of the same particles that are typically used in laboratory work.

In this work, second-harmonic generation (SHG) was utilized to probe, on the molecular-level, the change in the interfacial water degree of ordering next to silica surface being aged via multiple cycles of freezing and melting. We found that the



temperature dependent restructuring of water molecules is changing with the aging of the surface. Freezing/melting cycles enhance the phenomenon. The degrees of liquid water molecules ordering next to the surface at room temperature and shortly before freezing decrease and then increase again with sample aging. However, the water ordering–cooling dependence improves continuously with aging. This is accompanied with a continuous improvement in the IN ability of the surface. We interpret the observed changes in the surface properties in terms of dissolution and re-adsorption of the dissolution products at the sample surface.

## 2 Experimental

MilliQ water (18.2 MΩ·cm) of total organic content below 4 ppb was used in all experiments. The pH solutions were freshly prepared before the experiments. The high and low pH solutions were prepared from NaOH and HCl (Sigma Aldrich), respectively. The bulk pH was measured at room temperature using a pH meter (Orion 720A+, Thermo electronic corporation) and a pH electrode (ORION 8102 BN, ThermoFisher Scientific). The pH measurement set-up was calibrated using buffers with known pH values. The pH value is temperature dependent since the dissociation constant changes with temperature (Bandura and Lvov, 2006; Zumdahl, 1993). The measuring cell was sealed from the lab environment to avoid dissolved gases in the liquid solutions. The concentration of dissolved Si ions in the solution after experiment was measured using inductively coupled plasma mass spectrometry (ICP-MS, X-Series II, ThermoScientific). Fresh prism like UV fused silica sample (from Thorlabs GmbH) was cleaned first by soaking it in chloroform, acetone, and then ethanol (~ two hours each without sonication or 30 min with sonication, both giving the same result). Finally, the sample was flushed with MilliQ water. After each experiment series, the sample was cleaned with the same procedure but excluding the chloroform step. Before starting an experimental series (successive freezing / melting cycles), the sample was soaked in pH12 (NaOH) for 5 hours to generate fresh surface and finally in MQ water for another few hours and then washed with MQ water to remove excess sodium.

The SHG experiments were conducted using a 1 KHz femtosecond laser system (Solstice, Spectra Physics, 800 nm, 3.5 mJ, ~80 fs) with a beam diameter of ~2 mm at the interface. More details on the optical setup and the measuring cell can be found elsewhere (Abdelmonem et al., 2015; Abdelmonem et al., 2017; Abdelmonem, 2017). The fundamental beam was incident on the interface, Fig. 1, and the SHG signal was measured in co-propagating total internal reflection (TIR) geometry. A half-wave plate followed by a cube polarizer was used to adjust the polarization of the incident beam. The generated signal was filtered, using a 400 nm band pass filter, before being measured using a photomultiplier tube (PMT). The measured signal amplitude at certain polarization depends on the amount and structure of the interfacial molecules (Zhuang et al., 1999; Rao et al., 2003; Jang et al., 2013). It originates from the nonresonant electric dipolar contribution (Goh et al., 1988; Luca et al., 1995; Fordyce et al., 2001) and proportionate to the incident field and the second-order nonlinear susceptibility $\chi^{(2)}$ of the interface. For a charged interface, a third-order nonlinear polarization is induced by the static electric field due to the third-order nonlinear susceptibility $\chi^{(3)}$ of the solution (Ong et al., 1992; Zhao et al., 1993).

A silica prism was used as a sample where the hypotenuse was exposed to the solution. The incident angle of the fundamental beam was adjusted to 1° above the critical angle of TIR for silica-water interface to guarantee a TIR condition in the studied temperature range. In this work, the term Fresnel factors refers to nonlinear Fresnel factors affecting the SHG signal due to the optical constants of the media at the interface (Zhuang et al., 1999). The advantage of working close to the critical angle is that both PM (P-polarized SHG / 45°-polarized incident) and SM (S-polarized SHG / 45°-polarized incident) polarization combinations depend on only one non-vanishing nonlinear susceptibility tensor element ($\chi_{zzz}$) and ($\chi_{yyz}$), respectively (Shen, 1989; Zhuang et al., 1999). PM, however, gives a stronger signal and higher signal to noise ratio, and for this reason we only considered PM polarization. Nevertheless, we performed some runs with MP and MS polarizations combinations to assure that both give the same information (Fig. S2). The laser power coupled to the sample was 50 mW. The SHG signal is mainly produced by polarizable entities at the interface where the inversion symmetry is broken. Under




TIR geometry, the contribution of polarizable molecules next to the solid surface is limited by the penetration depth inside the contact medium (gas, liquid or ice). The calculated penetration depths for air, liquid water, and ice in the above described geometry are about 143 nm, 720 nm, and 414 nm respectively. The time resolution of the SHG measurements is 2.5 sec. The standard deviation of the measurements is less than 10%.

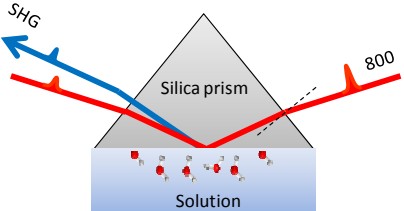

Figure 1: Sample and beams geometry (see text for details).

A cold stage (Linkam model HFS-X350) was used, in a homemade temperature-controlled environmental chamber, to apply a preset temperature profile (TP), and the SHG signal and substrate temperature were measured accordingly. In this work,

we are not seeking for the exact onset freezing temperature, but rather the qualitative change of the freezing efficiency and the structural behavior of interfacial water molecules before, during and after freezing. The silica prism was attached to the cold-stage and the surface of interest was exposed to the sample solution filled in a Teflon cell during the experiments. More details and drawing of the environmental chamber can be found elsewhere (Abdelmonem, 2017).

To allow qualitative comparisons, all results presented here were obtained using a standard TP, unless otherwise noted. For a

standard run, the cell was filled with 1mL volume of the solution of interest. In each run, the silica sample was kept in contact with the sample solution at 20 °C for 10 min, cooled down to -40 °C at a rate of 5 °C/min, held at -40 °C for 5 min, heated up to 0 °C at a cooling rate of 20 °C/min, held at 0 °C for 15 min to allow melting and departing of the ice from the surface region, and finally heated up to 20 °C at a rate of 20 °C/ min. This cooling profile was repeated successively to observe the changes in the freezing efficiency and water structure as a function of cooling cycle. The last two cycles were

delayed for some hours to assess possible aging of the sample in contact with solution in the absence of freezing/melting effects. The time of a complete standard TP is 45 min. In the following, the signal before freezing is labeled "liquid signal", the peak at the freezing event is termed "transient freezing peak", the signal after the freezing and transient freezing peak is labeled "ice signal", the peak at the melting event is defined as "transient melting peak", and the signal directly after the transient melting peak is termed the "confined liquid signal".




## 3 Results and discussion

Figure 2 shows the SHG signal observed for pH = 3 solution in contact with silica surface as a function of time during 25 TP cycles. We chose pH = 3 because it is close to the point of zero charge of the used sample and by that we eliminate the contribution from $\chi^{(3)}$. Also at this pH dissolution rate is minimum as mentioned in the introduction. Starting from 20 °C

5    and cooling down, the liquid signal in cycle 1 gradually decreases with temperature until the freezing point. At the freezing point, we observe a transient signal (a fast increase and subsequent decrease in the signal) upon freezing. After freezing and at constant temperature, the ice signal remains more or less flat. After melting we observe a high signal (confined liquid signal) which lasts for ~ 4 minutes. This period depends on the water volume and the holding time of ice at -40 °C. Finally, after heating back to 20 °C, we observe a liquid signal which is lower than the initial 20 °C liquid signal. When repeating the

10   TP 25 times, we observe clear changes in the SHG-temperature dependence. The gradual decrease in the liquid signal with cooling before the freezing event gradually transforms to an increase. The liquid signal versus cooling decreases for the early TP cycles, remains constant during the intermediate TP cycles, and then increases for the later TP cycles, as indicated by the green arrows in Fig. 3. A transient melting peak becomes visible from TP cycle 5. Finally, after melting started and after the associated transient melting peak, we observe a signal from liquid confined between the surface and the remaining ice

15   (confined liquid signal) the height of which decreases with the repeating TP cycles.

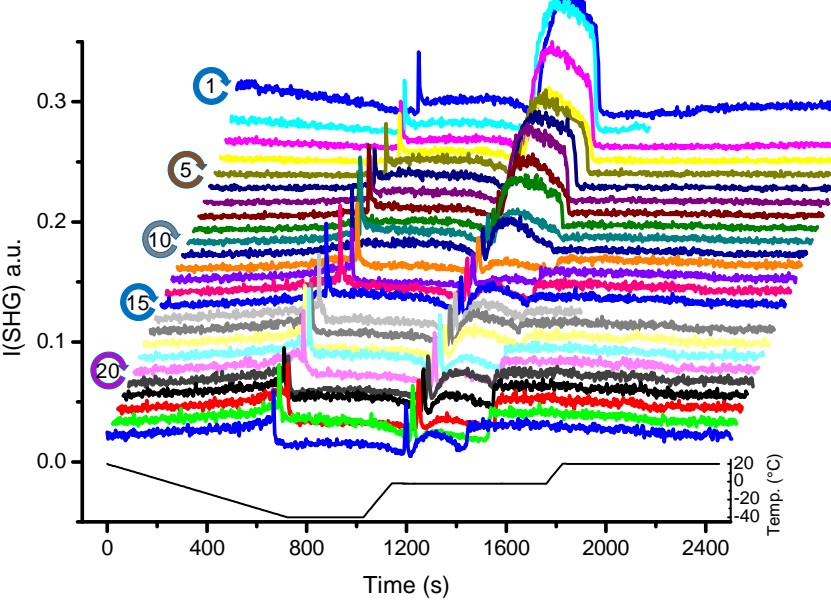

Figure 2: SHG signal at pH = 3 solution-silica interface as a function of time for 25 successive TP cycles



### 3.1 Liquid signal

In Figs. 2 and 3, the SHG signal from time zero to the transient freezing peaks show the liquid signal as a function of time / temperature during cooling from 20 °C to the freezing event. This part of the TP indicates a restructuring of interfacial water as a function of temperature before freezing. In cycle 1, the liquid signal is relatively high at 20 °C (comparatively strongly ordered water molecules) and decreases with cooling indicating decreasing order in the water structure with decreasing temperature. This signal-temperature dependency weakens with repeating the TP. In cycle 5, the liquid signal remains almost constant during cooling. In cycle 10, the liquid signal starts to slightly increase with cooling, particularly before the freezing event. Finally, cycle 25 shows a significant increase in the water signal with cooling indicating that the surface induces a higher degree of water molecules ordering upon cooling.

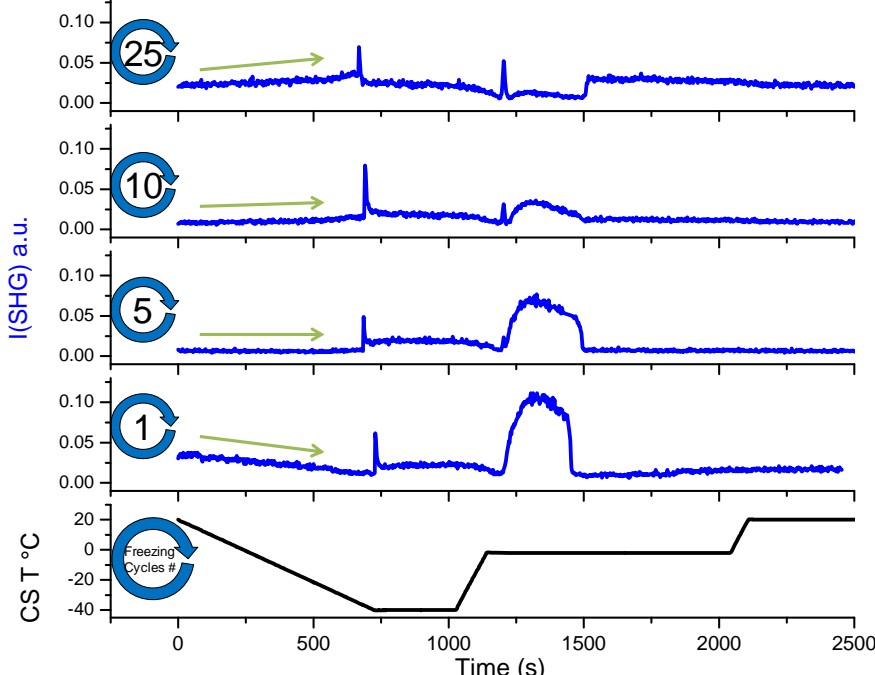

Figure 3: A selection from the set of plots shown in Fig. 2. This set of subplots shows clear change in the I(SHG ) – temperature dependence during the TP iteration. CS T= Cold-stage temperature

To gain more insight into the change in the water restructuring with temperature upon repeated TP cycles, we plot the SHG intensity as a function of cooling cycle at different temperatures. Figure 4a, b and c show the liquid signal as a function of TP cycle number at the beginning of each cycle (at 20 °C), shortly before the freezing event (at -31 °C), and immediately before the freezing event, respectively. The temperature of the latter slightly differs for each cycle. Figure 4a shows a relatively high liquid signal at 20 °C in the first cycle. The 20 °C signal decreases and then increases again showing a minimum, located at approximately the 7[th] TP cycle, indicating a minimum degree of water molecules ordering at the interface in this cycle.



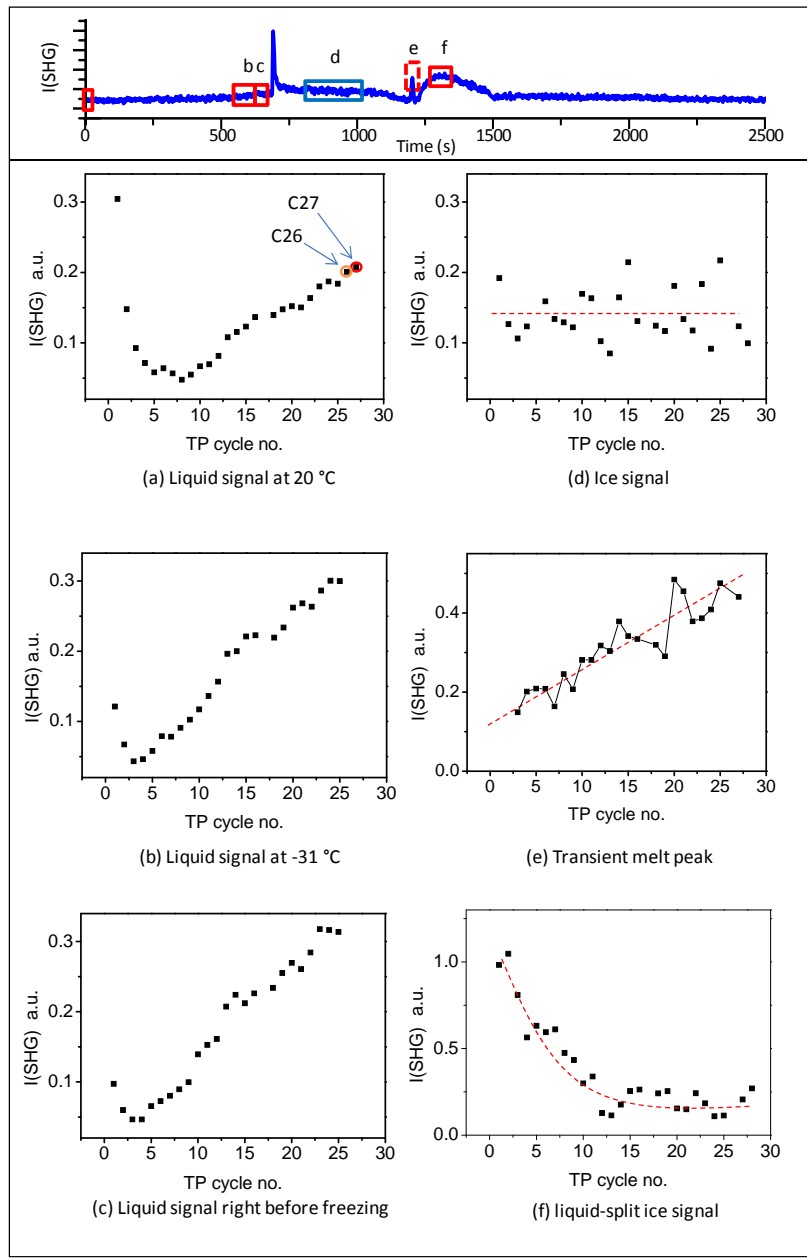

Figure 4: Upper panel: a sample plot of SHG vs. time during the standard TP. Lower panel: The averaged SHG signal as a function of TP cycle number at the different time slots marked with red and blue rectangles in the upper panel. The red rectangle corresponds to liquid phase while the blue rectangle corresponds to ice phase. The dashed red lines on the plots are guiding lines through the data points.



The cycles from 1 to 25 have been carried out with equal time intervals. However, to assure that the changes we observe here are mainly due to the freezing/melting effect and not simply time dependent surface aging in contact with pH = 3 solution, we carried out two additional cycles after deliberately waiting for 5 hours while the sample was in contact with pH = 3 at room temperature. This time is equivalent to the time needed to perform about 6 TP cycles. The 20 °C liquid signals of these

two additional cycles are marked as orange (C26) and red (C27) circles in Fig. 4a. C26 has the same standard TP of the preceding 25 cycles while the red C27 has a different TP for a reason to be discussed later. By comparing the plot of the SHG signal vs. cooling cycle number (Fig. 4a), and vs. time (Fig. S3), we find that the two data points of the two additional cycles (C26 and C27) fit well with the extrapolation of the changes in the SHG signals with TP cycle number. However, these two points are offset in terms of time when extrapolating the SHG signal with time. We conclude that the changes in

the surface–solution system are accelerated by the freezing and melting processes. It is well established that silica surfaces dissolve in aquatic environments (Hiemstra and van Riemsdijk, 1990). As mentioned in the introduction, dissolution is accelerated by very low and high pH values. The amount of aqueous silica depends on the pH and the time of exposure. However, the observed changes in the 20 °C SHG signal as a function of TP cycle number, Fig. 4a, within the relatively short time intervals, about 45 min per cycle, can only be attributed to an accelerated dissolution upon freezing/melting

followed by re-adsorption of dissolution products, hence changing the surface properties which in turn influences the interfacial water structure.

The decrease and subsequent increase in the liquid signal at 20 °C with the progress of the TP cycles, Fig. 4a, can be explained as follows: At the beginning, before the minimum point, adsorbed dissolution products interrupt the water structure which is expected to be slightly ordered due to the H-bonding or the weak surface charge at pH = 3. As mentioned

above, pH = 3 is close to the point of zero charge of silica and therefore the signal is low compared to neutral or high pH, Fig. S4. Each freezing/melting cycle increases the concentration of the dissolution products in the bulk solution sufficiently to change the ionic strength of the interfacial water and this in turns affects the interfacial water structure. At the minimum point, adsorbate concentration at the surface causes the highest disorder of the water molecules. Beyond the minimum point, the adsorbate concentration is sufficiently high to induce a certain degree of arrangement of water molecules. This results in

an increase of the liquid signal at 20 °C beyond the minimum point. The alteration of the signal trend originates from an interplay between screening and interference effects (Schaefer et al., 2017). In the following, the interaction before the minimum point will be termed a "screening phase" and that after minimum an "interference phase".

Shortly before the freezing event the liquid signal as a function of TP cycle number also shows a minimum as can be seen in the Fig. 4b and c, respectively. Figure 4b shows the SHG liquid signal at -31 °C while Fig. 4c shows the signal immediately

before the freezing point, for all TP cycles. Figures 4b and c show similar behavior. In both cases the minimum point occurs between cycles 3 and 4, i.e. not at the same position as the minimum point of the 20 °C liquid signal (around cycle 7). This suggests that temperature affects the adsorption-desorption balance. The effect of re-adsorption, on water structure before the minimum point, at lower temperature precedes that at higher temperature. This means that cooling favors the uptake of dissolved silica (i.e. adsorption) and hence increases the interfacial ion concentration. Considering the thermodynamic

behavior of silica in such systems (see SI, Fig. S1) the solubility of amorphous silica strongly decreases with decreasing temperature. Adsorption of inorganic solutes to mineral oxide surfaces is typically strongly related to the solubility behavior (Lützenkirchen and Behra, 1995) so that the thermodynamic calculations in SI support the idea of dissolved silica enhances the re-adsorption to the fused silica surface under cooling. This justifies the hypotheses that the change in the liquid signal with an increasing number of TP cycles is due to the uptake of dissolved silica ions. The decrease in liquid signal with

temperature decrease within the screening phase is consistent with the conclusion that in this phase re-adsorption causes disorder of water molecules and that cooling accelerates the completion of this phase.

Beyond the minimum, the older the sample the higher the degree of ordering induced by the surface on the interfacial water particularly at supercooled condition. In addition, the liquid signal-temperature dependence per TP cycle converts from





inversely to directly proportional with cooling (green arrows in Fig. 3). It has been shown in different studies on water-mineral oxide interface that higher degree of ordering of liquid water molecules next to the surface is an indication of higher heterogeneous freezing efficiency (Abdelmonem, 2017; Abdelmonem et al., 2017; Abdelmonem et al., 2015; Anim-Danso et al., 2013; Anim-Danso et al., 2016; Yang et al., 2011). In the present work, it was our intention to impose a constant cooling

rate to qualitatively follow the freezing efficiency and interfacial water structural changes under identical conditions. Figure 5 shows the freezing temperatures obtained under the standard TP and experimental conditions described above. The overall trend shows that the older the sample the higher the temperature of freezing, i.e. the earlier the freezing event, under the same conditions. This means that indeed the aged sample exhibits better IN efficiency which is induced by the pre-structuring of water molecules by the modified surface.

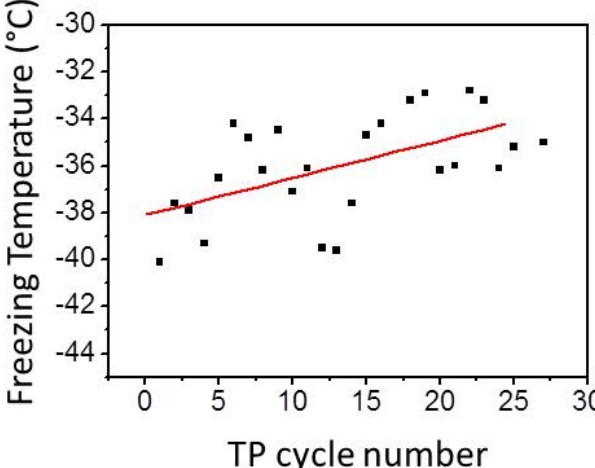

Figure 5: Freezing temperature of pH = 3 solution as a function of the TP cycle number under the experimental conditions mentioned in the text. Black squares represent experimental data. Red line is a linear fit to follow the trend of the data points.

For quantification we need to find the range of bulk concentrations at which we may observe alteration in the signal
behavior. As a control experiment, we prepared dissolved silica solutions with different concentrations of silicic acid at pH = 3 and measured the SHG signal at the interface for each solution at room temperature. Figure 6 shows the SHG intensity as a function of ion concentration. Indeed, we observe oscillations in the signal as the silica concentration increase. Without invoking any model, this directly supports the conclusion that the change in the SHG signal in our freezing/melting experiments was due to the change in the dissolution products concentration with the progress of the TP cycles. The
minimum we see in Fig. 4a should correspond to one of the minima in the silica ions concentration experiment, Fig. 6. By analyzing the pH = 3 solution in contact with the silica surface after the complete set of TP cycles, we found that the aqueous concentration of Si was $41.0 \pm 1.5$ μM. This means that by repeating the freezing/melting TP cycle the dissolution products generated a substantial interfacial ionic strength equivalent to that at the low concentration range of Fig. 6 (shadowed area). The dissolution of silica changed the z-dependent electrostatic potential of the bare silica/water interface in a very similar
manner as the addition of approximately 41μM silicic acid at pH = 3 at room temperature, implying an interfacial concentration of dissolution-released entities of the same order. This control experiment corroborates the scenario of multi adsorption phases of dissolution products at the surface and the concomitant effect it has on the water structure.

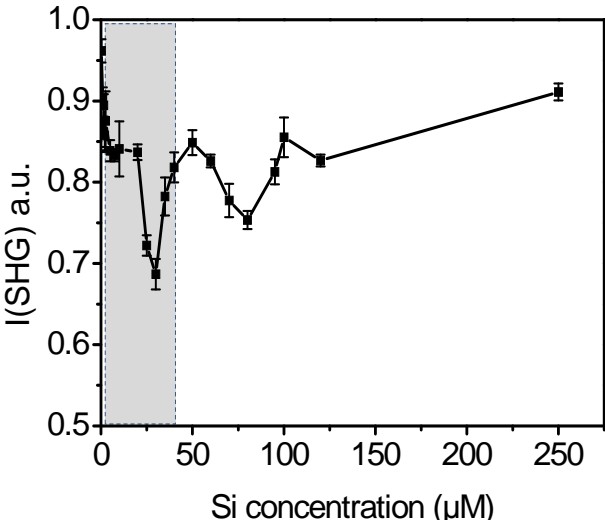

Figure 6: SHG intensity as a function of silica ions concentration at pH = 3 at 20 °C. The error bars represent the standard deviation from the corresponding average value.

**3.2 Transient freezing and melting peaks**

Although this study was not aimed at discussing the transient freezing peak, the obtained data enforce us to reconsider this recently observed ambiguous phenomenon. The transient peak upon freezing was reported at pH9.8 for the aqueous solution-mica interface using SFG spectroscopy (Anim-Danso et al., 2016), for a neutral water-silica interface using SFG spectroscopy (Lovering et al., 2017), for a neutral water-mica and -sapphire interface using SHG spectroscopy (Abdelmonem, 2017), and for a pH9 aqueous solution-sapphire interface using SFG spectroscopy (Abdelmonem et al.,

2018). Anim-Danso et al. supposed that the transient signal is due to progressive events occurring near the surface during the phase transition without specifying potential process. Lovering et al. suggested the presence of a transient stacking-disordered ice at the interfaces during freezing. Abdelmonem et al. (2018) have reported that the observed transient signals arise from a smooth transition between water and ice and does not necessarily indicate transient species. It was demonstrated that the transient change in the signal intensity results from an interference between different SFG peak parameters changing

at different rates. All SFG and SHG studies mentioned above showed a transient increase in the signal, although with different temporal characteristics (few tens of seconds to several minutes).

In the present study we adjudicate on this debate and provide evidence that the transient freezing peak arises from the multi-interface problem, and the reported observations may indeed be explicable without the need to invoke any transient species. We also address the parameters affecting the transient time. In our experiments we observed the transient signal at freezing

and melting as well. To interpret the transient signal we corrected our data for the Fresnel factors. Figure 7 shows the corrected and non-corrected SHG signal around the transient freezing and transient melting peaks of TP cycle number 25. The liquid data before the transient freezing peak and after the transient melting peak are corrected to the water Fresnel factors (red lines) and the ice data between the transient freezing peak and the transient melting peak are corrected to the ice Fresnel factors (blue lines). All corrected data sets have then been normalized to the measured ice signal for the sake of

clearness. For this reason the Fresnel corrected ice data (blue lines) coincide with the non-Fresnel corrected signal (black solid circles). The lower panels of Fig. 7 show a schematic representation of the development of the ice forming and melting at the interface around the freezing and melting events, respectively, involving six steps: 1. Before the freezing event, the SHG signal arises from the liquid-solid interface and is affected by the Fresnel factors of bulk liquid and bulk solid (silica). 2. Once the freezing occurs, a thin film of ice is formed at the surface. This film is sufficiently thin for the ice-solid interface





SHG signal to be still affected by the Fresnel factors of bulk liquid and bulk solid. 3. After the freezing event, the ice film grows and the SHG signal is generated from ice-solid interface and affected by the Fresnel factors of bulk ice and bulk solid. 4. Before the melting event (similar to 3). 5. Once the melting occurs, a thin film of liquid is formed at the surface. This film is sufficiently thin so that the SHG signal at liquid-solid interface is still affected by the Fresnel factors of bulk ice and bulk

solid. 6. Later after the melting event, the ice melts further and the SHG signal is generated from liquid-solid interface and affected by the Fresnel factors of bulk liquid and bulk solid.

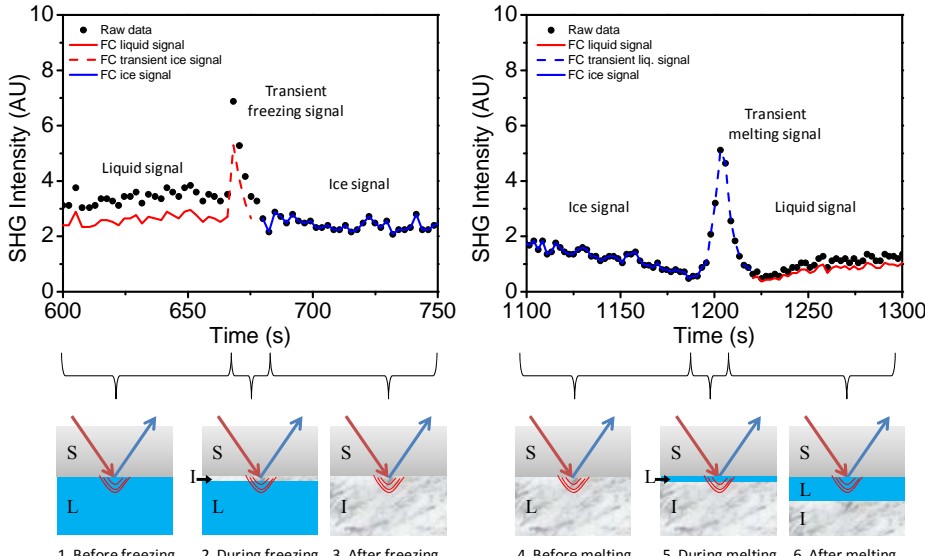

Figure 7: Upper panel: A comparison between SHG signal, corrected (continuous and dashed lines) and non-corrected (black solid circles) to the Fresnel factors, around the transient freezing peak (left panel) and around the transient melting peak (right panel) for TP cycle
number 25. Red lines correspond to liquid phase, blue lines correspond to ice phase, and dashed lines correspond to transient signal. Lower panel: A schematic representation of the development of the ice forming and melting at the interface around freezing and melting events, respectively.

The transient signals persist until the respective ice or liquid film thickness exceeds the evanescent field (represented by the red bows at the interfaces in the lower panel of Fig. 7). Considering the transient freezing, if we assume that the bulk liquid

optical constants are the origin of the transient ice signal, then correcting the transient ice signal to liquid-solid Fresnel factors should lead to a maximum not higher than the ice signal, but this is not the case. Correcting the transient ice signal to the liquid-solid Fresnel factors gives the dashed red line in Fig. 7 which has a maximum significantly higher than the ice signal. Similar justification applies to the transient melting if we correct the transient liquid signal to the ice-solid Fresnel factors (dashed blue line in Fig. 7). Thus, the Fresnel Factor corrections cannot exclusively explain the transient freezing

behavior.

We conclude here that the effect of the Fresnel Factors, or in other words the optical constants of the bulk isotropic media around the interface, is not sufficient to generate the observed transient peak. However, another multi-interface problem is accompanying the formation and growth of ice and liquid films after the freezing and melting events, respectively. In the case of freezing, once the IN starts and until the thickness of the ice film exceeds the depth of the evanescent field, two

interfaces are involved and can contribute directly to the signal: the ice-solid interface and the liquid-ice interface (see drawing 2 in the lower panel of Fig. 7). The SHG signal of the first interface (ice-solid) should be comparable to that of the ice signal (blue line, upper left panel, Fig. 7). The SHG signal of the second interface (liquid-ice), however, is an additional contribution to the detected signal which emerges for a very short time (i.e. the time of vertical growth of the ice film within the evanescent field). In the case of melting, before the thickness of the liquid film exceeds the depth of the evanescent field,





(drawing 5, lower panel, Fig. 7) an additional temporary signal from the ice-liquid interface is contributing to the liquid-solid interface. Again, the ice-liquid signal should be sufficiently higher than that of the liquid-solid interface at pH = 3 and will last for short time until the liquid layer thickness exceeds the evanescent field. The signal generated at the secondary interface has not been considered in the above discussed SFG and SHG studies

We also believe that this transient signal occurs in all similar systems, while its detection depends on data acquisition speed and the geometry of the measuring cell. The data represented here were collected at time resolution of 2.5 sec. If the time resolution would be greater than 9 sec, which is the average duration of the transient freezing peaks under our standard TP, a transient freezing signal would not be observable for this system. The transient time depends on the experimental conditions of the system, e.g. the cooling rate, the sample volume and the thermal conductivities of the two isotropic media. At the

melting event, a thin water film is formed between solid and ice which starts to grow vertically at the expense of the ice melting (drawing 5, lower panel, Fig. 7). A quasi liquid layer has been reported on the ice surface in both atmospheric ice and multiphase chemistry as well as in commercial contexts. The growth of such a film is, however, slower than the growth of the ice film after freezing, because the energy transfer between solid surface and ice is slower than that between the solid surface and water. The vertical growth of the thin water film formed upon melting is then expected to last for a relatively

longer time than that of the ice sheet after the freezing event. Indeed, the average transient melting time was about 16 seconds in our standard TP (cycles from 1 to 25). We tested the heat transfer effect in one very slow cooling profile (TP of C27) which is shown in Fig. S5. The very slow cooling evolved enough time for the bulk liquid to cool down, enhancing by this the vertical ice growth rate compared to the standard TP used for runs 1 to 25. Once the freezing occurred in C27, the propagation of the ice in the z direction was so fast that the 2.5 sec resolution of the detection system was not able to detect

the transient freezing signal. Although no transient freezing temperature was observed in C27, the transient melting peak was detected for this TP cycle and was even longer (~ 48 sec) than for the first 25 cycles of the standard TP. This is because the longer cooling time developed a larger ice bulk in the measuring cell, i.e., a bigger heat absorber than the bulk ice in the standard TP cycles. This generates more resistance to the growth of the thin water film.

    In summary, we conclude that the transient SHG/SFG signals observed in the above mentioned studies were a result of the

multi-interface problem. However, using such a signal under controlled conditions can allow for measuring crystal growth which is so far a challenging task due to the incomplete understanding of the physical mechanisms underlying ice crystallization. It is worth to notice here that the height of the transient melting peak is proportional to the sample aging caused by successive TP cycles and, in our system, starts after cycle 3, Fig. 4e. This corroborates the role of the interference phase after the minimum point as discussed in the liquid signal section. We have not attempted to plot the transit freezing

peak height in Fig. 4. Due to the limited detection time resolution and the fast change in the transient freezing signal, the peak region involved only one data point from which the actual peak height cannot be determined. With this we presented a reasonable interpretation of the transient signal which was under debate in the recent years.

### 3.3 Ice signal

    The ice signal, Fig. 4d, did not show any clear trend with repeating the TP. However we noticed strong fluctuations in the

level of the averaged ice signal after the freezing.

### 3.4 Confined liquid signal

    The confined liquid signal, labelled (f) in the upper panel of Fig. 4, is generated at the interface between the silica surface and bulk liquid confined between the silica surface and ice after melting a relatively thick interfacial layer (drawing 6 lower panel, Fig. 7). This confined bulk liquid is sufficiently thick for the evanescent field not to reach the bulk ice. It is hence a

signal only generated at the liquid-solid interface. This signal is strongly influenced by the geometry of our experiment and can change from one cell design to another (see Fig. S6). In our cell design, the ice piece which separates from the sample





surface after melting stays close to the surface and requires some time to be detached from the interfacial region and move towards the water not been frozen during the experiment (see Fig. S6). Although this piece of ice does not contribute directly to the signal, it indirectly influences the SHG signal as a result of two simultaneous effects: 1. Depending on the distance between this ice piece and the surface, the orientation of the interfacial water may be influenced by the ice surface.

2. The melting of this ice piece slowly dilutes the solution back to pH = 3. The combination of these two simultaneous processes result in the hump we see in the SHG curves, Fig. 2 and 3, in the region after the transient melting peak. A supplementary video, shows the development of this hump with the melting and motion of this ice piece from after the melting event to complete melting, is available (Abdelmonem et al., 2019). The height of the confined liquid signal decreases with repeating the TP, Fig. 4f. This is an additional evidence that the surface-solution system is significantly

changing with the iteration of the TP. Regardless of the vague behavior of the confined liquid signal, optimizing this setup may ultimately allow us to extend our studies to investigate interfacial water melted between ice and solid surface or the so called quasi-liquid layer (Nagata et al., 2019; Li and Somorjai, 2007; Rosenberg, 2005; Döppenschmidt et al., 1998) which receives strong interest from different industrial applications (e.g. ski sports, frozen food packaging,...).

**4 Conclusions**

The effect of surface aging under acidic conditions on the rearrangement of interfacial water molecules next to an amorphous silica surface has been studied at the molecular level using SHG spectroscopy under temperature-controlled conditions. The aging has been accelerated by successive freezing / melting cycles of a pH = 3 solution next to the surface. Similar aging is highly probable for mineral oxide aerosols in the atmosphere. The aging altered the temperature dependence of the water structure next to the surface and correspondingly the ice nucleation ability of the surface. The alteration in the water-silica

interaction could be understood in terms of the disruption of the equilibrium at the surface due to dissolution and re-adsorption of dissolution products. The re-adsorption of dissolved silica generates a network on the surface. The SHG intensity versus freeze-melt cycle number at constant temperature indicated alteration in the generated network. A control experiment showed an oscillation in the degree of order of water molecules with Si ion concentration. The first minimum in this oscillation quantified the dissolution-released entities in the aging experiments. Beside the main findings, we gave a

reasoning explanation of the transient freezing and melting signals which were recently observed by SFG and SHG studies, on similar systems, and were questioned in terms of reality and origin. The results provide new insights in our understanding of the consequences of surface aging in aqueous solutions and under atmospheric conditions. This study is expected to benefit future atmospheric research, particularly cloud formation and aerosol aging in atmosphere, with potential implications for the pharmaceutical and food industries.


ASSOCIATED CONTENT

Supporting Information. The supporting information comprises thermodynamic considerations, and associated data and plots. This material is available free of charge via the Internet at.

AUTHOR INFORMATION

Corresponding Authors

E-mail: ahmed.abdelmonem@kit.edu,

Conflicts of interest

There are no conflicts to declare.

ACKNOWLEDGMENTS





AA is grateful to the German Research Foundation (DFG, AB 604/1-1,2). SR thanks the DAAD (2017/18, 57299294). The authors are grateful to Mischa Bonn for useful discussions and acknowledging Horst Geckeis, Thomas Leisner, Frank Heberling, Dieter Schield and Teba Gil-Diaz for their support.

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
