# Peer review of "Cloud history can change water-ice-surface interactions of oxide mineral aerosols: a case study on silica"

_Atmospheric Chemistry and Physics, 2019_

## Referee Comment (RC1) · Anonymous Referee #1 · 12 Nov 2019

The manuscript submitted by Abdelmonem et al. examines the effects of freeze-melt processes on the aqueous chemistry at silica surfaces at low pH. The experiments were performed in an environmental cell in conjunction with second-harmonic generation spectroscopy. Abdelmonem et al. found a water ordering-cooling dependence that improved continuously and they proposed that water ordering is a result of the dissolution of the silica surface and that this process causes the improved ice nucleation of aged silica samples. The manuscript is interesting but focused mostly on SHG measurements. Therefore it seems more relevant for a more physical chemical journal and in the current form the conclusions of the SHG measurements and their relevance and connection for atmospheric research are not evident.

Scientific significance:  3
Scientific quality:  2
Presentation quality: 3

**Points to be addressed:**

**General discussion:**
SHG probes the average of the ensemble of all water molecules underneath the heterogeneous silica sample. Yet, ice nucleation on α-quartz surfaces was shown to occur at only a few locations which were associated with micron-size surface pits. (Holden et al. 2019 Science Advances). The authors should include this important point into the manuscript so that the readers can better put the SHG results in perspective. The authors should also think about possible topographical changes on the silica due to pH 3 treatment as this might also be able to explain the improved ice nucleation. The authors might also want to consider providing evidence whether their silica substrates become better ice nucleators with time.

**Introduction:**
p.3. l. 5- The discussion about the pH solubility of silica is not presented in great clarity and it is not directly clear why pH 3 is relevant for atmospheric conditions. When reading the introduction it seems that higher alkaline pH would be much more relevant.

p.3 l. 42 "molecular level" this statement should be altered. In the presented experiments a large ensemble of water molecules are probed the term molecular-level might be misleading.

**Experimental:**

p.4 l. 15 Are there any information on the surface roughness or homogeneity of your silica samples?

p.4. l. 34    "the incident angle was adjusted to 1 degree above the critical anlge of TIR to guarantee a TIR condition in the studied temperature range.  Why was +1 degree chosen and why would the incident angle change?  Which effect would the changing angle of incident have on your data?

p.4 l. 35  Were the Fresnel factors corrected for the effect of temperature? Does it affect the silica measurements?

**Results and discussion:**

Does the volume stay the same during the longer successive runs and are evaporation effects possible/considered?

p. 6 Fig. 2: The light grey scan and the turquoise scan were cut after 1500 s without an explanation. Full data sets should be shown.

p.8 Fig.4: Fig4b and c seem identical. It would be much more convincing if they would add a point at 2000 s to see how this signal changes as a function of cycles. From the data presented in Figure 2 it seems that except for Run 1 the intensities look comparable.

p.8 Fig. 4 Are there any indications from your data that the onset of the freezing occurs at earlier timings when the samples are aged?

p.9 l. 1-5 Control experiments at lower temperature, not RT would be helpful since the pH depends on the temperature.

p.9 l. 41 The statement "the older the sample" is somewhat ambiguous. Were the experiments performed on multiple independent samples? Or one silica prism and the age of the sample refers to the number of cycles the sample was exposed to. The number of used silica samples and the number of independent experiments should be added to the materials section.

p.10 l. 8 What is the experimental evidence that the prism that orders water better is also a better ice nucleator? Were any experiments performed? Couldn't the ageing process and the contact with acid also just roughen the surface and that is the reason why it nucleates better?

p.10 Fig.5 Since the observed effect is not very pronounced. It would be good toadd error bars in the Figure or provide information on how reproducible the trend is.

p.11 l. 15 The results of the following study should be added to the discussion. Rehl et al. 2019 New Insights into $\chi(3)$ Measurements: Comparing Nonresonant Second Harmonic Generation and Resonant Sum Frequency Generation at the Silica/Aqueous Electrolyte Interface, JPCA.

p.13 l. 33 Is there an explanation why the ice signal shows such strong variations.

p. 13. l. 35 Is it possible to estimate the thickness of the liquid film? I would assume this could provide very useful information as one could estimate the pH of this solution which should be much lower due to the freeze concentration and likely dissolves the silica even faster.

p. 14 l. 28 "this study is expected to benefit"… The connection of the results of the current study and its implications for atmospheric research are not clear.

---

## Referee Comment (RC2) · Anonymous Referee #2 · 19 Nov 2019

The manuscript reports the effect of freeze-melt cycles on water freezing at silica surfaces. The experimental data represent some new and interesting phenomena. It will take the community years to fully understand these phenomena. This reviewer recommends its publication with some minor revisions.

Suggested revisions: (1) In Fig. 2, 3, and 7, it could be more intuitive to use the temperature as the x axis, instead of the time. (2) The data presented in the manuscript may not directly related to "cloud history", as indicated by the title.

---

## Author Comment (AC1) · 6 Dec 2019

*Point-to-point answers to the comments of Referee #1*

*The authors would like to thank the referee for his/her time reading the manuscript and placing the comments and suggestions. We believe that considering the points suggested by the Referee has improved the manuscript significantly.*

RC: 1)

The manuscript submitted by Abdelmonem et al. examines the effects of freeze-melt processes on the aqueous chemistry at silica surfaces at low pH. The experiments were performed in an environmental cell in conjunction with second-harmonic generation spectroscopy. Abdelmonem et al. found a water ordering-cooling dependence that improved continuously and they proposed that water ordering is a result of the dissolution of the silica surface and that this process causes the improved ice nucleation of aged silica samples. The manuscript is interesting but focused mostly on SHG measurements. Therefore it seems more relevant for a more physical chemical journal and in the current form the conclusions of the SHG measurements and their relevance and connection for atmospheric research are not evident.

*AC1:*

*The authors generally agree with the opinion of the Reviewer, but would like to clarify important points:*

*SHG has only been used here as a technique to examine the water structure at the surface. The authors didn't add novelty in SHG theory or discuss its details. ACP has published many articles on ice nucleation and interactions in atmosphere using ESM, AFM, IR-spectroscopy …etc. All these articles were considered atmosphere relevant based on the aim of the study irrespective of the technique. Indeed, the application of SHG and SFG in atmospheric science is something new and beneficial for atmospheric science and the respective journals to consider new techniques.*

*This work has been triggered by the wide scatter of experimental results on ice nucleation abilities of atmospheric aerosol particles, as has been well demonstrated by (Hoose and Mohler, 2012) in their ACP review of results from six decades of laboratory experiments of heterogeneous ice nucleation. Hoose and Mohler concluded with the recommendation of performing experiments (spectroscopic, microscopic and chemical characterization methods) with pure and homogeneous materials to improve the understanding of the basic physical and chemical principles of heterogeneous ice nucleation. This manuscript discusses the influence of surface aging on heterogeneous freezing, a crucial process in cloud formation. Therefore the authors believe that the Journal of "Atmospheric Chemistry and Physics" is the most relevant.*

RC: 2)

SHG probes the average of the ensemble of all water molecules underneath the heterogeneous silica sample. Yet, ice nucleation on α-quartz surfaces was shown to occur at only a few locations which were associated with micron-size surface pits. (Holden et al. 2019 Science Advances). The authors should include this important point into the manuscript so that the readers can better put the SHG results in perspective. The authors should also think about possible topographical changes on the silica due to pH 3 treatment as this might also be able to explain the improved ice nucleation. The authors might also want to consider providing evidence whether their silica substrates become better ice nucleators with time.

*AC2:*

*The authors would like to point that the sample is not heterogeneous but rather a homogeneous polished fused silica sample. This manuscript doesn't touch on active sites, rather the role of surface interactions in enhancing or resisting the ice formation. Our sample is fused silica which is not*

*identical to α-quartz. Nevertheless, the authors have cited the work suggested by the Reviewer as it demonstrates the difficulty of predicting and controlling crystal formation.*

*The authors couldn't detect any topographical changes on the silica due to pH 3 treatment. AFM measurements yielded no results that would support such change. Figure (AC1.1) shows a comparison between (1) cleaned sample and (b) sample after experiment. The authors did not rely on these results as a proof that the topography has not been changed because the AFM measurements were done off-line and hitting the exact spot where SHG was measuring was not guaranteed.*

[Figure]

*Figure (AC1.1): AFM measurements on the surface of the silica sample after cleaning (a) and after freezing-melting experiments (b).*

*We see the increase in the in freezing temperature under identical conditions as a function of aging, Figure (5) in the manuscript. High statistics freezing assay could provide second evidence that silica substrates become better ice nucleators, but it was not available during this study.*

**Introduction:**

RC: 3)

p.3. l. 5- The discussion about the pH solubility of silica is not presented in great clarity and it is not directly clear why pH 3 is relevant for atmospheric conditions. When reading the introduction it seems that higher alkaline pH would be much more relevant.

*AC3:*

*In the introduction, the high pH was only an example on the effect of droplet size change on the pH value of the cloud droplet. This is also valid for low pH. Slightly acidic or alkaline will be extreme acidic or alkaline with droplet evaporation, respectively. As mentioned in the manuscript, silica dissolution in aquatic solution is always possible however is extreme at high and very low pH as well. The choice of pH3 in the spectroscopic measurements has different reasons. As mentioned in the p.5. l.2-3, "We chose pH = 3 because it is close to the point of zero charge of the used sample and by that we eliminate the contribution from $\chi^{(3)}$. Also at this pH the dissolution rate is minimum as mentioned in the introduction."*

RC: 4)

p.3 l. 42 "molecular level" this statement should be altered. In the presented experiments a large ensemble of water molecules are probed the term molecular-level might be misleading.

*AC4:*

*The "molecular level" term is associated with SHG and SFG since their first results provided by Shen (Shen, 1989), as the signal allows in-situ mapping of the molecular arrangement at the surface. We only re-use the term, it is not our definition. The SHG/SFG signal depends on the "molecular hyperpolarizability tensor elements" which could allow the determination of absolute orientation of the molecules at the surface (Goh et al., 1988). We add here a selection of SHG/SFG articles which directly used the term "molecular level":*

*(Silva and Miranda, 2016; Gonella et al., 2016; Jerome et al., 2002; Lambrakos et al., 1998; Xiao et al., 1997; Tohda, 1996; Ashwell et al., 1992; Wang et al., 2019; Schlegel et al., 2019; Li et al., 2019; Feugmo et al., 2019; Ulrich et al., 2017; Takeshita et al., 2017; Abdelmonem et al., 2017a; Zhang et al., 2016; Wang et al., 2016; Leng et al., 2016; Luca et al., 1995).*

**Experimental:**

RC: 5)

p.4 l. 15 Are there any information on the surface roughness or homogeneity of your silica samples?

*AC5:*

*The authors agree with the Reviewer, the surface specifications were missing in the manuscript. The surface of the silica samples is optically polished, surface quality is 40-20 Scratch-Dig and surface flatness at 633 nm is Lambda/10. This information has been added to the experimental section in the revised manuscript.*

RC: 6)

p.4. l. 34 "the incident angle was adjusted to 1 degree above the critical anlge of TIR to guarantee a TIR condition in the studied temperature range. Why was +1 degree chosen and why would the incident angle change? Which effect would the changing angle of incident have on your data?

*AC6:*

*The issue is more about the change in the critical angle and not the incident angle. The critical angle for TIR at the silica/water interface is a function of the refractive indices of both silica and water. It is well known that the refractive index is temperature and phase-of-matter dependent. We fix the incident angle, but the critical angle may cross the incident angle during cooling and freezing and then violates the TIR condition. Violation of the TIR condition will result in a clear drop in the signal and make data interpretation difficult. Any incident angle higher than the range of changes in the critical angle with temperature should be acceptable. Figure (AC1.2) shows the change in the refractive indices of water and ice and the critical angle with temperature in the temperature range of our experiments. Simple calculations based on Snell's law show that the change in the critical angle for the water/silica system in the studied temperature range (red triangles) is less than 0.5°. We used an incident angle = critical angle + 1°, however any higher angle would also be fine regarding the TIR condition.*

[Figure]

*Figure AC1.2: The change in the Refractive index and the corresponding change in critical angle of TIR as a function of temperature for Si/water interface. The temperature dependent refractive indices are taken from (Cho et al., 2001; Waxler and Cleek, 1971)*

RC: 7)

p.4 l. 35 Were the Fresnel factors corrected for the effect of temperature? Does it affect the silica measurements?

*AC7:*

*We did not correct the data in Figure (4) for the effect of temperature. The effect of temperature on Fresnel factors in the range of temperatures studied here is small for the water/silica interface, (red crosses, Figure AC1.3). However this is not the reason to ignore them. As each panel in Figure 4 shows the change in SHG with aging at constant temperature, correction to Fresnel factors has no effect and this is the reason why data in Fig(4) are not Fresnel corrected.*

*Only the discussion on the transient signal, Figure 7, required Fresnel factors correction, for two reasons: 1) Data are compared at both different temperatures and different phase-of-matter (liquid and ice). 2) There is a significant difference between Fresnel factors of liquid and ice as can be seen in Figure AC1.3 (red crosses and blue X respectively).*

[Figure]

*Figure AC1.3: The change in the Fresnel factors as a function of temperature for silica/water interface (red crosses) and the value for Si/ice interface (blue X).*

RC: 8)

Does the volume stay the same during the longer successive runs and are evaporation effects possible/considered?

*AC8:*

*The measuring cell is tightly closed and sealed during the measurements. Volume changes are not expected.*

RC: 9)

p. 6 Fig. 2: The light grey scan and the turquoise scan were cut after 1500 s without an explanation. Full data sets should be shown.

*AC9:*

*We thank the Reviewer for this remark. We have included a comment concerning this in the figure caption in the revised version. Sometimes, due technical reasons, the data acquisition software crashes and some data points are missing, (e.g. the light grey (cycle 2) and the turquoise (cycle 16) scans). The lost data in these two scans are in the end of the scan where, as can be seen from the other 23 scans, no exceptional change is expected. Omitting cycles 2 and 16 from the presented data set wouldn't affect our interpretations. However, we wanted to present all collected scans with constant interval between them.*

RC: 10)

p.8 Fig.4: Fig4b and c seem identical. It would be much more convincing if they would add a point at 2000 s to see how this signal changes as a function of cycles. From the data presented in Figure 2 it seems that except for Run 1 the intensities look comparable.

*AC10:*

*Indeed Fig 4c seems identical to 4b. However, it is our intention to show that at the onset point, Fig 4c, nothing exceptional happens although the onset temperatures are different. Nevertheless it is a good point to show the change in signal with aging at other temperatures. Since the paper discusses the restructuring of water upon cooling and relates this to the freezing process, data at 2000 s may not be the right choice. At time 2000 s there is liquid signal after partial melting with undefined amounts of melted ice and solute concentration. What could be useful to compare after melting is the liquid signal at room temperature after each complete freeze-melt cycle. But this is already included in panel a (e.g. the room temperature liquid signal after first scan is the room temperature liquid signal of the second cycle).*

*We select here a set of temperatures during cooling to plot the liquid signal as a function of cycle number. Figure AC1.4 shows the averaged SHG liquid signal as a function of TP cycle number at five different temperatures on the cooling path. The minimum points occur at lower cycle numbers for lower temperatures (summarized in table AC1.1). This supports our conclusion that cooling favors the uptake of dissolved silica (i.e. adsorption).*

[Figure]

| Time with respect to scan start (s) | Temperature (°C) | Closest cycle no. to minimum signal |
|---|---|---|
| a) 0 sec | 20 | 7 |
| b) 120 sec | 10 | 6 |
| c) 240 sec | 0 | 5 |
| d) 480 sec | -20 | 4 |
| e) 630 sec | -32 | 3 |

*Table AC1.1: The selected temperatures in Figure AC1.4 and the closest TP cycle number of minimum SHG liquid signal.*

*Figure AC1.4 and Table AC1.1 have been included in the supporting information and the corresponding discussion has been changed in the revised manuscript.*

*Figure AC1.4: Upper panel: a sample plot of SHG vs. time/temperature during cooling. Lower panel: The averaged SHG liquid signal as a function of TP cycle number at five different temperatures during cooling before freezing. The red lines on the plots are guiding lines through the data points.*

RC: 11)

p.8 Fig. 4 Are there any indications from your data that the onset of the freezing occurs at earlier timings when the samples are aged?

*AC11:*

*Yes this is directly indicated by Figure 5.*

RC: 12)

p.9 l. 1-5 Control experiments at lower temperature, not RT would be helpful since the pH depends on the temperature.

*AC12:*

*The control experiments included the full temperature range. Comparing lower temperature shows the same result: i.e. "Pausing the freeze-melt cycles for 5 hours has minimal effect on the SHG signal in time", as can be seen in Figure AC1.5. We have commented on this point in the discussion in the revised manuscript and replaced Figure S3 by Figure AC1.5. Please note that comparing time axis in case of C27 is only possible at RT (20 °C) because C27 has different cooling rate as described in the manuscript.*

[Figure]

*Figure AC1.5: SHG signal at pH3 solution-silica interface as a function of TP cycle number (a and c) and time (b and d) of liquid signal at 20 °C (a and b) and -31 °C (c and d) during repeating the freeze-melt TP. The dashed red lines are trend lines illustrating the behavior of the signal. For low temperature (-31 °C), only data point C26 is plotted because C27 has a different cooling rate as described in the manuscript. CS26 and CS27 are lay on the trend line with plot against TP cycle number (a and c) but not with time (b and c). This shows that the significant aging we observe in this work arises from the freeze-melt process and not from the time the sample is in contact with solution.*

RC: 13)

p.9 l. 41 The statement "the older the sample" is somewhat ambiguous. Were the experiments performed on multiple independent samples? Or one silica prism and the age of the sample refers to the number of cycles the sample was exposed to. The number of used silica samples and the number of independent experiments should be added to the materials section.

*AC13:*

*We thank the reviewer for pointing to this misleading term. For consistency, the data reported in this work were all collected on the same silica prism. This statement has been added to the Experimental section in the revised version.*

*The age of the sample refers to the number of cycles the sample was exposed to. The statement "the older the sample…" has been corrected in the revised version to "the older the surface, i.e. the more often exposed to freeze-melt cycles, the higher the degree of …"*

RC: 14)

p.10 l. 8 What is the experimental evidence that the prism that orders water better is also a better ice nucleator? Were any experiments performed? Couldn't the ageing process and the contact with acid also just roughen the surface and that is the reason why it nucleates better?

*AC14:*

*This is a good question and we have clarified this point in the revised version.*

*The experimental evidence that a surface that orders water better "could be" a better ice nucleator has been recently reported (Abdelmonem et al., 2017b; Abdelmonem et al., 2015; Yang et al., 2011). Other parameters, e.g. roughness, porosity, steps…, are not excluded. We believe that all surface properties influence the ice nucleation ability though with different weights. We also believe that the way these properties influence the interaction with water molecules at the surface is the key to the overall effect. In our previous work (Abdelmonem et al., 2017a) we combined freezing assays and SFG characterizations to study the effect of surface charge on the heterogeneous ice-nucleation ability of α-alumina (0001) surfaces. We are not generalizing our former observation on that particular surface (i.e. the α-alumina (0001)), but only recall an existence of evidence.*

*That "Water ordering leads to better ice nucleation condition" is not that straight forward. When a surface is able to create an ordering compatible with the structure of a crystalline phase, it will then promote the nucleation of the corresponding phase particularly if the induced ordering further matches the crystalline structure at a higher degree (Bi et al., 2017). A surface may exhibit the ordering patterns that resemble the structure of ice. Therefore, water layers bound to surfaces may be ice-like, providing a template for ice to nucleate (Bi et al., 2016). Based on these facts we suggest that the re-adsorbed dissolution products have an arrangement on its surface as close as possible to that of water molecules in some low index plane of ice.*

*There is a lack in the literature on what happens at the surface on the molecular level and our approach is applied in ice nucleation studies only since few years ago. With this work we try to attract the attention of theoreticians who can simulate this re-adsorption of dissolution products and their arrangement on the surface. Indeed we have proposed MD simulations with the group of Molecular Modelling and Computer Simulations, Clemson University, as future cooperation.*

RC: 15)

p.10 Fig.5 Since the observed effect is not very pronounced. It would be good toadd error bars in the Figure or provide information on how reproducible the trend is.

*AC15:*

*The authors agree with the Reviewer, this information was missing in the figure caption. Repeating the experiment showed the same trend over the whole range of cycles with an average standard deviation of 1.3. The information has been added in the figure caption with the corresponding error bars on Figure 5 in the revised version of the manuscript.*

RC: 16)

p.11 l. 15 The results of the following study should be added to the discussion. Rehl et al. 2019 New Insights into χ(3) Measurements: Comparing Nonresonant Second Harmonic Generation and Resonant Sum Frequency Generation at the Silica/Aqueous Electrolyte Interface, JPCA.

*AC16:*

*The study of Rehl et al. 2019, although interesting, is not relevant to our study. Rehl et al. compare the SHG and SFG from technical point of view. Neither the sample (IR-grade Silica) nor the solution (0.5 M NaCl) are the same as ours.  Even the temperature effect was not discussed. They wanted to find the origin of inconsistencies between SHG and SFG that have arisen when comparing experiments on silica at high electrolyte concentrations. Discussing their results in our manuscript will confuse the reader and deviate from the main study.*

*However, this paper includes one very useful information which is "SHG is more sensitive to the number density of aligned water, particularly at low pH and ionic strength".  As already mentioned in the first paragraph in the "Results and Discussion" section, we eliminate the $\chi^{(3)}$ effect by choosing pH = 3, and we also add no salts. We have cited this paper in this context in the revised manuscript.*

RC: 17)

p.13 l. 33 Is there an explanation why the ice signal shows such strong variations.

*AC17:*

*So far we have no explanation for the strong fluctuation in the ice signal after freezing. This will need further investigations on the ice structure after freezing. One expected scenario is the formation of free OH group after freezing (as we observed on Sapphire 110 surface using SFG, not published yet). The role of the formation of this free OH is not yet known. We added the following sentence to p.13. l. 33 "This ice signal fluctuation merits further investigations particularly using SFG which gives details on the individual contributions of different interfacial species from their resonant vibrations.". We also merged the "Ice signal" and "Confined liquid signal" sessions.*

RC: 18)

p. 13. l. 35 Is it possible to estimate the thickness of the liquid film? I would assume this could provide very useful information as one could estimate the pH of this solution which should be much lower due to the freeze concentration and likely dissolves the silica even faster.

*AC18:*

*Not with this technique. The maximum penetration depth in our geometry is ~ 400 nm. The layer thickness exceeds this distance very shortly after the freezing point (= transient peak time).*

RC: 18)

p. 14 l. 28 "this study is expected to benefit"… The connection of the results of the current study and its implications for atmospheric research are not clear.

*AC19:*

*The authors have answered on this question in AC1 (second paragraph)*

References:

Abdelmonem, A., Lützenkirchen, J., and Leisner, T.: Probing Ice-Nucleation Processes on the Molecular Level using Second Harmonic Generation Spectroscopy, Atmos. Meas. Tech., 8, 3519-3526, doi: 10.5194/amt-8-3519-2015, 2015.

Abdelmonem, A., Backus, E. H. G., Hoffmann, N., Sanchez, M. A., Cyran, J. D., Kiselev, A., and Bonn, M.: Surface-charge-induced orientation of interfacial water suppresses heterogeneous ice nucleation on alpha-alumina (0001), Atmospheric Chemistry and Physics, 17, 7827-7837, doi: 10.5194/acp-17-7827-2017, 2017a.

Abdelmonem, A., Backus, E. H. G., Hoffmann, N., Sánchez, M. A., Cyran, J. D., Kiselev, A., and Bonn, M.: Surface-Charge-Induced Orientation of Interfacial Water Suppresses Heterogeneous Ice Nucleation on α-Alumina (0001), Atmos. Chem. Phys., 17, 7827-7837, doi: 10.5194/acp-17-7827-2017, 2017b.

Ashwell, G. J., Hargreaves, R. C., Baldwin, C. E., Bahra, G. S., and Brown, C. R.: IMPROVED 2ND-HARMONIC GENERATION FROM LANGMUIR-BLODGETT-FILMS OF HEMICYANINE DYES, Nature, 357, 393-395, doi: 10.1038/357393a0, 1992.

Bi, Y., Cabriolu, R., and Li, T.: Heterogeneous Ice Nucleation Controlled by the Coupling of Surface Crystallinity and Surface Hydrophilicity, The Journal of Physical Chemistry C, 120, 1507-1514, doi: 10.1021/acs.jpcc.5b09740, 2016.

Bi, Y., Cao, B., and Li, T.: Enhanced heterogeneous ice nucleation by special surface geometry, Nature Communications, 8, 15372, doi: 10.1038/ncomms15372, 2017.

Cho, C. H., Urquidi, J., Gellene, G. I., and Robinson, G. W.: Mixture model description of the T-, P dependence of the refractive index of water, The Journal of Chemical Physics, 114, 3157-3162, doi: 10.1063/1.1331571, 2001.

Feugmo, C. G. T., Liegeois, V., Caudano, Y., Cecchet, F., and Champagne, B.: Probing alkylsilane molecular structure on amorphous silica surfaces by sum frequency generation vibrational spectroscopy: First-principles calculations, Journal of Chemical Physics, 150, doi: 074703

10.1063/1.5080007, 2019.

Goh, M. C., Hicks, J. M., Kemnitz, K., Pinto, G. R., Heinz, T. F., Eisenthal, K. B., and Bhattacharyya, K.: Absolute orientation of water molecules at the neat water surface, J. Phys. Chem., 92, 5074-5075, doi: 10.1021/j100329a003, 1988.

Gonella, G., Lutgebaucks, C., de Beer, A. G. F., and Roke, S.: Second Harmonic and Sum-Frequency Generation from Aqueous Interfaces Is Modulated by Interference, Journal of Physical Chemistry C, 120, 9165-9173, doi: 10.1021/acs.jpcc.5b12453, 2016.

Hoose, C. and Mohler, O.: Heterogeneous ice nucleation on atmospheric aerosols: a review of results from laboratory experiments, Atmos. Chem. Phys., 12, 9817-9854, doi: 10.5194/acp-12-9817-2012, 2012.

Jerome, B., Schuddeboom, P. C., and Meister, R.: Rotational friction at the molecular level, Europhysics Letters, 57, 389-395, doi: 10.1209/epl/i2002-00473-7, 2002.

Lambrakos, S. G., Trzaskoma-Paulette, P. P., and Triandaf, I. A.: Surface-site charge-displacement model for analysis of electromodulated optical second-harmonic response at a metal surface, Applied Spectroscopy, 52, 1240-1247, doi: 10.1366/0003702981945066, 1998.

Leng, C., Sun, S. W., Zhang, K. X., Jiang, S. Y., and Chen, Z.: Molecular level studies on interfacial hydration of zwitterionic and other antifouling polymers in situ, Acta Biomaterialia, 40, 6-15, doi: 10.1016/j.actbio.2016.02.030, 2016.

Li, X., Ma, L., and Lu, X. L.: Interfacial Molecular-level Structures of Polymers and Biomacromolecules Revealed via Sum Frequency Generation Vibrational Spectroscopy, Jove-Journal of Visualized Experiments, doi: 10.3791/59380, 2019.

Luca, A. A. T., Hebert, P., Brevet, P. F., and Girault, H. H.: Surface second-harmonic generation at air/solvent and solvent/solvent interfaces, J. Chem. Soc. Faraday Trans., 91, 1763-1768, doi: 10.1039/ft9959101763, 1995.

Schlegel, S. J., Hosseinpour, S., Gebhard, M., Devi, A., Bonn, M., and Backus, E. H. G.: How water flips at charged titanium dioxide: an SFG-study on the water-TiO2 interface, Physical Chemistry Chemical Physics, 21, 8956-8964, doi: 10.1039/c9cp01131e, 2019.

Shen, Y. R.: Surface Properties Probed by Second-Harmonic and Sum-Frequency Generation, Nature, 337, 519-525, doi: 10.1038/337519a0, 1989.

Silva, H. S. and Miranda, P. B.: Probing the Molecular Ordering and Thermal Stability of Azopolymer Layer-by-Layer Films by Second-Harmonic Generation, Langmuir, 32, 9950-9959, doi: 10.1021/acs.langmuir.6b02486, 2016.

Takeshita, N., Okuno, M., and Ishibashi, T.: Molecular conformation of DPPC phospholipid Langmuir and Langmuir-Blodgett monolayers studied by heterodyne-detected vibrational sum frequency generation spectroscopy, Physical Chemistry Chemical Physics, 19, 2060-2066, doi: 10.1039/c6cp07800a, 2017.

Tohda, K.: Studies on the mechanism of the potential generation at the surface of ion-selective liquid membranes at the molecular level, Bunseki Kagaku, 45, 641-657, 1996.

Ulrich, N. W., Li, X., Myers, J. N., Williamson, J., Lu, X. L., and Chen, Z.: Distinct Molecular Structures of Edge and Middle Positions of Plasma Treated Covered Polymer Film Surfaces Relevant in the Microelectronics Industry, Ieee Transactions on Components Packaging and Manufacturing Technology, 7, 1377-1390, doi: 10.1109/tcpmt.2017.2718562, 2017.

Wang, F., Li, X., Zhang, F. R., Liu, X. Y., Hu, P. C., Beke-Somfai, T., and Lu, X. L.: Revealing Interfacial Lipid Hydrolysis Catalyzed by Phospholipase A(1) at Molecular Level via Sum Frequency Generation Vibrational Spectroscopy and Fluorescence Microscopy, Langmuir, 35, 12831-12838, doi: 10.1021/acs.langmuir.9b02284, 2019.

Wang, M. C., Li, B. L., Chen, Z., and Lu, X. L.: Molecular-Level Structures at Poly(4-vinyl pyridine)/Acid Interfaces Probed by Nonlinear Vibrational Spectroscopy, Journal of Polymer Science Part B-Polymer Physics, 54, 848-852, doi: 10.1002/polb.23978, 2016.

Waxler, R. M. and Cleek, G. W.: Refractive indices of fused silica at low temperatures, J. Res. Natl. Inst. Stand. Technol., 75A, 279-281, doi: 10.6028/jres.075A.026, 1971.

Xiao, C. Y., Feng, J. K., Huang, X. R., Jia, Q., Sun, J. Z., Fang, Q., and Jiang, M. H.: A theoretical study on the second harmonic generation properties of molecules and crystals of 4,5-bis(2',4'-dinitrophenylthio)-1,3-dithiole-2-one(BNPT-DTO) and 4,5-bis(2',4'-dinitrophenylthio)-1,3-dithiole-2-thione(BNPT-DTT), Acta Chimica Sinica, 55, 866-871, 1997.

Yang, Z., Bertram, A. K., and Chou, K. C.: Why Do Sulfuric Acid Coatings Influence the Ice Nucleation Properties of Mineral Dust Particles in the Atmosphere?, J. Phys. Chem. Lett., 2, 1232-1236, doi: 10.1021/jz2003342, 2011.

Zhang, X. X., Myers, J. N., Huang, H., Shobha, H., Chen, Z., and Grill, A.: SFG analysis of the molecular structures at the surfaces and buried interfaces of PECVD ultralow-dielectric constant pSiCOH, Journal of Applied Physics, 119, doi: 10.1063/1.4942442, 2016.

---

## Author Comment (AC2) · 6 Dec 2019

*__Point-to-point answers to the comments of Referee #2__*

The authors would like to thank the referee for his/her time reading the manuscript and placing the comments. We also acknowledge his positive feedback and valuable suggestions.

RC: 1)

In Fig. 2, 3, and 7, it could be more intuitive to use the temperature as the x axis, instead of the time.

*AC1:*

*The authors would like to thank the Reviewer for his practical suggestion. Indeed plotting the data as a function of temperature will be more perceptive. However, this is only possible in the range of scan where the independent parameter (temperature in this case) is single-valued, otherwise the plot will be confusing, e.g. Fig. AC2.1.*

*Since we focus in this manuscript on the water restructuring upon cooling, plotting the SHG liquid signal as a function of temperature during cooling is the correct choice. Since Figure 3 is a subset of Figure 2, we have only replaced Figure 3 with a new one that now includes temperature as the x axis (Figure AC2.2).*

[Figure]

*Figure AC2.1: This figure shows haw Figure 3 would look like if we plot the SHG signal of the complete scans with temperature is the x-axis.*

[Figure]

*Figure AC2.2: SHG liquid signal as a function of time and temperature during cooling. Figure 3 in the manuscript has been replaced with this Figure.*

*Figure 7 cannot be plotted using temperature as the x axis because the temperature around the freezing and melting peaks is almost constant (i.e. again a not single-valued relation).*

RC: 2)

The data presented in the manuscript may not directly related to "cloud history", as indicated by the title.

*AC2:*

*As mentioned in the abstract and introduction, an aerosol-containing cloud droplet can go through different freeze-melt or evaporation-condensation\* cycles, so that not only the aerosol surface structure may change, but also ionic strength and pH of the cloud droplet. We conclude that the cloud history may affect the contained aerosol particles. However, we agree with the Reviewer that the title of the original manuscript is may be improved. The new title is being discussed between the authors and will finally be decided after finalizing the reviewed manuscript version.*

*\* During cloud formation, water vapor condenses on aerosol particles forming liquid suspended water droplets in about a 100% RH environment. Cloud droplets are constantly forming and dissipating. Depending on the atmospheric conditions (e.g. temperature, RH, or air draft) the cloud droplet change its size. In case of temperature increase, cloud mixing with drier air, or air sinking within the cloud, cloud droplets may evaporate (may also totally dissipate). Under these conditions the acidic, basic components or ionic strength in the cloud droplet will reach extreme values. We show in this manuscript that this may significantly change the surface properties of mineral oxide aerosols.*

---

## Author Response (AR1)

**Cloud history can change water-ice-surface interactions of oxide mineral aerosols: a case study on silica**

Ahmed Abdelmonem[1*], Sanduni Ratnayake[2], Jonathan D. Toner[3] and Johannes Lützenkirchen[2]

[1]Institute of Meteorology and Climate Research - Atmospheric Aerosol Research (IMKAAF), Karlsruhe Institute of Technology (KIT), 76344 Eggenstein-Leopoldshafen, Germany
[2]Institute of Nuclear Waste Disposal (INE), Karlsruhe Institute of Technology (KIT), 76344 Eggenstein-Leopoldshafen, Germany
[3]Department of Earth & Space Sciences, University of Washington, Seattle, WA 98195, USA
* Correspondence to: A. Abdelmonem (ahmed.abdelmonem@kit.edu)

*Correspondence to*: **Ahmed Abdelmonem (ahmed.abdelmonem@kit.edu)**

**Final Rebuttal**
**I. Point-to-point response to Reviewer 1**

**II. Point-to-point response to Reviewer 2**

**III. Revised manuscript with tracked changes**

**IV. Revised SI with tracked changes**

Message from the Authors:

The authors would like to thank the referees for their time reading the manuscript and placing the comments and suggestions. We believe that considering the points suggested by the Referees has improved the manuscript significantly. In addition we have improved the presentation quality.

**I. Point-to-point response to Referee 1 (RC1)**

RC1: 1)

The manuscript submitted by Abdelmonem et al. examines the effects of freeze-melt processes on the aqueous chemistry at silica surfaces at low pH. The experiments were performed in an environmental cell in conjunction with second-harmonic generation spectroscopy. Abdelmonem et al. found a water ordering-cooling dependence that improved continuously and they proposed that water ordering is a result of the dissolution of the silica surface and that this process causes the improved ice nucleation of aged silica samples. The manuscript is interesting but focused mostly on SHG measurements. Therefore it seems more relevant for a more physical chemical journal and in the current form the conclusions of the SHG measurements and their relevance and connection for atmospheric research are not evident.

*AC1:1)*

*The authors generally agree with the opinion of the Reviewer, but would like to clarify important points:*

*SHG has only been used here as a technique to examine the water structure at the surface. The authors didn't add novelty in SHG theory or discuss its details. ACP has published many articles on ice nucleation and interactions in atmosphere using ESM, AFM, IR-spectroscopy …etc. All these articles were considered atmosphere relevant based on the aim of the study irrespective of the technique. Indeed, the application of SHG and SFG in atmospheric science is something new and beneficial for atmospheric science and the respective journals to consider new techniques.*

*This work has been triggered by the wide scatter of experimental results on ice nucleation abilities of atmospheric aerosol particles, as has been well demonstrated by (Hoose and Mohler, 2012) in their ACP review of results from six decades of laboratory experiments of heterogeneous ice nucleation. Hoose and Mohler concluded with the recommendation of performing experiments (spectroscopic, microscopic and chemical characterization methods) with pure and homogeneous materials to improve the understanding of the basic physical and chemical principles of heterogeneous ice nucleation. This manuscript discusses the influence of surface aging on heterogeneous freezing, a crucial process in cloud formation. Therefore the authors believe that the Journal of "Atmospheric Chemistry and Physics" is the most relevant.*

RC1: 2)

SHG probes the average of the ensemble of all water molecules underneath the heterogeneous silica sample. Yet, ice nucleation on α-quartz surfaces was shown to occur at only a few locations which were associated with micron-size surface pits. (Holden et al. 2019 Science Advances). The authors should include this important point into the manuscript so that the readers can better put the SHG results in perspective. The authors should also think about possible topographical changes on the silica due to pH 3 treatment as this might also be able to explain the improved ice nucleation. The authors might also want to consider providing evidence whether their silica substrates become better ice nucleators with time.

*AC1:2)*

*The authors would like to point that the sample is not heterogeneous but rather a homogeneous polished fused silica sample. This manuscript doesn't touch on active sites, rather the role of surface interactions in enhancing or resisting the ice formation. Our sample is fused silica which is not identical to α-quartz. Nevertheless, the authors have cited the work suggested by the Reviewer as it demonstrates the difficulty of predicting and controlling crystal formation.*

*The authors couldn't detect any topographical changes on the silica due to pH 3 treatment. AFM measurements yielded no results that would support such change. Figure (AC1.1) shows a comparison between (1) cleaned sample and (b) sample after experiment. The authors did not rely on these results as a proof that the topography has not been changed because the AFM measurements were done off-line and hitting the exact spot where SHG was measuring was not guaranteed.*

[Figure]

*Figure (AC1.1): AFM measurements on the surface of the silica sample after cleaning (a) and after freezing-melting experiments (b).*

*We see the increase in the freezing temperature under identical conditions as a function of aging, Figure (5) in the manuscript. High statistics freezing assay could provide second evidence that silica substrates become better ice nucleators, but it was not available during this study.*

**Introduction:**

RC1: 3)

p.3. l. 5- The discussion about the pH solubility of silica is not presented in great clarity and it is not directly clear why pH 3 is relevant for atmospheric conditions. When reading the introduction it seems that higher alkaline pH would be much more relevant.

*AC1:3)*

*In the introduction, the high pH was only an example on the effect of droplet size change on the pH value of the cloud droplet. This is also valid for low pH. Slightly acidic or alkaline will be extreme acidic or alkaline with droplet evaporation, respectively. As mentioned in the manuscript, silica dissolution in aquatic solution is always possible however is extreme at high and very low pH as well. The choice of pH3 in the spectroscopic measurements has different reasons. As mentioned in the original manuscript p.5. l.2-3, "We chose pH = 3 because it is close to the point of zero charge of the used sample and by that we eliminate the contribution from $\chi^{(3)}$. Also at this pH the dissolution rate is minimum as mentioned in the introduction."*

RC1: 4)

p.3 l. 42 "molecular level" this statement should be altered. In the presented experiments a large ensemble of water molecules are probed the term molecular-level might be misleading.

*AC1:4)*

*The "molecular level" term is associated with SHG and SFG since their first results provided by Shen (Shen, 1989), as the signal allows in-situ mapping of the molecular arrangement at the surface. We only re-use the term, it is not our definition. The SHG/SFG signal depends on the "molecular hyperpolarizability tensor elements" which could allow the determination of absolute orientation of the molecules at the surface (Goh et al., 1988). We add here a selection of SHG/SFG articles which directly used the term "molecular level":*

*(Silva and Miranda, 2016; Gonella et al., 2016; Jerome et al., 2002; Lambrakos et al., 1998; Xiao et al., 1997; Tohda, 1996; Ashwell et al., 1992; Wang et al., 2019; Schlegel et al., 2019; Li et al., 2019; Feugmo et al., 2019; Ulrich et al., 2017; Takeshita et al., 2017; Abdelmonem et al., 2017a; Zhang et al., 2016; Wang et al., 2016; Leng et al., 2016; Luca et al., 1995).*

**Experimental:**

RC1: 5)

p.4 l. 15 Are there any information on the surface roughness or homogeneity of your silica samples?

*AC1:5)*

*The authors agree with the Reviewer, the surface specifications were missing in the manuscript. The surface of the silica samples is optically polished, surface quality is 40-20 Scratch-Dig and surface flatness at 633 nm is Lambda/10. This information has been added to the experimental section in the revised manuscript (p. 4,l. 10-11).*

RC1: 6)

 p.4. l. 34 "the incident angle was adjusted to 1 degree above the critical anlge of TIR to guarantee a TIR condition in the studied temperature range. Why was +1 degree chosen and why would the incident angle change? Which effect would the changing angle of incident have on your data?

*AC1:6)*

*The issue is more about the change in the critical angle and not the incident angle. The critical angle for TIR at the silica/water interface is a function of the refractive indices of both silica and water. It is well known that the refractive index is temperature and phase-of-matter dependent. We fix the incident angle, but the critical angle may cross the incident angle during cooling and freezing and then violates the TIR condition. Violation of the TIR condition will result in a clear drop in the signal and make data interpretation difficult. Any incident angle higher than the range of changes in the critical angle with temperature should be acceptable. Figure (AC1.2) shows the change in the refractive indices of water and ice and the critical angle with temperature in the temperature range of our experiments. Simple calculations based on Snell's law show that the change in the critical angle for the water/silica system in the studied temperature range (red triangles) is less than 0.5°. We used an incident angle = critical angle + 1°, however any higher angle would also be fine regarding the TIR condition.*

[Figure]

*Figure AC1.2: The change in the Refractive index and the corresponding change in critical angle of TIR as a function of temperature for Si/water interface. The temperature dependent refractive indices are taken from (Cho et al., 2001; Waxler and Cleek, 1971)*

RC1: 7)

 p.4 l. 35 Were the Fresnel factors corrected for the effect of temperature? Does it affect the silica measurements?

*AC1:7)*

*We did not correct the data in Figure (4) for the effect of temperature. The effect of temperature on Fresnel factors in the range of temperatures studied here is small for the water/silica interface, (red crosses, Figure AC1.3). However this is not the reason to ignore them. As each panel in Figure 4 shows the change in SHG with aging at constant temperature, correction to Fresnel factors has no effect and this is the reason why data in Fig(4) are not Fresnel corrected.*

*Only the discussion on the transient signal, Figure 7, required Fresnel factors correction, for two reasons: 1) Data are compared at both different temperatures and different phase-of-matter (liquid and ice). 2) There is a significant difference between Fresnel factors of liquid and ice as can be seen in Figure AC1.3 (red crosses and blue X respectively).*

[Figure]

*Figure AC1.3: The change in the Fresnel factors as a function of temperature for silica/water interface (red crosses) and the value for Si/ice interface (blue X).*

RC1: 8)

Does the volume stay the same during the longer successive runs and are evaporation effects possible/considered?

*AC1:8)*

*As mentioned in the Experimental section, the measuring cell is tightly closed and sealed during the measurements. Volume changes are not expected.*

RC1: 9)

p. 6 Fig. 2: The light grey scan and the turquoise scan were cut after 1500 s without an explanation. Full data sets should be shown.

*AC1:9)*

*We thank the Reviewer for this remark. We have included a comment on this deficiency in the figure caption in the revised version. Sometimes, due technical reasons, the data acquisition software crashes and some data points are missing, (e.g. the light grey (cycle 2) and the turquoise (cycle 16) scans). The lost data in these two scans are in the end of the scan where, as can be seen from the other 23 scans, no exceptional change is expected. Omitting cycles 2 and 16 from the presented data set wouldn't affect our interpretations. However, we wanted to present all collected scans with constant interval between them.*

RC1: 10)

p.8 Fig.4: Fig4b and c seem identical. It would be much more convincing if they would add a point at 2000 s to see how this signal changes as a function of cycles. From the data presented in Figure 2 it seems that except for Run 1 the intensities look comparable.

*AC1:10)*

*Indeed Fig 4c seems identical to 4b. However, it is our intention to show that at the onset point, Fig 4c, nothing exceptional happens although the onset temperatures are different. Nevertheless it is a good point to show the change in signal with aging at other temperatures. Since the paper discusses the restructuring of water upon cooling and relates this to the freezing process, data at 2000 s may not be the right choice. At time 2000 s there is liquid signal after partial melting with undefined amounts of melted ice and solute concentration. What could be useful to compare after melting is the liquid signal at room temperature after each complete freeze-melt cycle. But this is already included in panel a (e.g. the room temperature liquid signal after first scan is the room temperature liquid signal of the second cycle).*

*We select here a set of temperatures during cooling to plot the liquid signal as a function of cycle number. Figure AC1.4 shows the averaged SHG liquid signal as a function of TP cycle number at five different temperatures on the cooling path. The minimum points occur at lower cycle numbers for lower temperatures (summarized in table AC1.1). This supports our conclusion that cooling favors the uptake of dissolved silica (i.e. adsorption).*

[Figure]

[Figure]

| Time with respect to scan start (s) | Temperature (°C) | Closest cycle no. to minimum signal |
|---|---|---|
| a) 0 sec | 20 | 7 |
| b) 120 sec | 10 | 6 |
| c) 240 sec | 0 | 5 |
| d) 480 sec | -20 | 4 |
| e) 630 sec | -32 | 3 |

*Table AC1.1: The selected temperatures in Figure AC1.4 and the closest TP cycle number of minimum SHG liquid signal.*

*Figure AC1.4 and Table AC1.1 have been included in the supporting information and the corresponding discussion has been changed in the revised manuscript.*

*Figure AC1.4: Upper panel: a sample plot of SHG vs. time/temperature during cooling. Lower panel: The averaged SHG liquid signal as a function of TP cycle number at five different temperatures during cooling before freezing. The red lines on the plots are guiding lines through the data points.*

RC1: 11)

p.8 Fig. 4 Are there any indications from your data that the onset of the freezing occurs at earlier timings when the samples are aged?

*AC1:11)*

*Yes this is directly indicated by Figure 5.*

RC1: 12)

p.9 l. 1-5 Control experiments at lower temperature, not RT would be helpful since the pH depends on the temperature.

*AC1:12)*

*The control experiments included the full temperature range. Comparing lower temperature shows the same result: i.e. "Pausing the freeze-melt cycles for 5 hours has minimal effect on the SHG signal in time", as can be seen in Figure AC1.5. We have replaced Figure S3 by Figure AC1.5 (now S4 in the revised version). The paragraph p. 9, l.1-8 has been changed correspondingly. Please note that comparing time axis in case of C27 is only possible at RT (20 °C) because C27 has different cooling rate as described in the manuscript.*

[Figure]

*Figure AC1.5: SHG signal at pH3 solution-silica interface as a function of TP cycle number (a and c) and time (b and d) of liquid signal at 20 °C (a and b) and -31 °C (c and d) during repeating the freeze-melt TP. The dashed red lines are trend lines illustrating the behavior of the signal. For low temperature (-31 °C), only data point C26 is plotted because C27 has a different cooling rate as described in the manuscript. CS26 and CS27 lay on the trend line with plot against TP cycle number (a and c) but not with time (b and c). This shows that the significant aging we observe in this work arises from the freeze-melt process and not from the time the sample is in contact with solution.*

RC1: 13)

p.9 l. 41 The statement "the older the sample" is somewhat ambiguous. Were the experiments performed on multiple independent samples? Or one silica prism and the age of the sample refers to the number of cycles the sample was exposed to. The number of used silica samples and the number of independent experiments should be added to the materials section.

*AC1:13)*

*We thank the reviewer for pointing to this misleading term. For consistency, the data reported in this work were all collected on the same silica prism. This statement has been added to the Experimental section in the revised version (p.4, l. 10-11).*

*The age of the sample refers to the number of cycles the sample was exposed to. The statement "the older the sample…" has been corrected in the revised version to "the older the surface, i.e. the more often exposed to freeze-melt cycles, the higher the degree of …" (p.9, l. 42)*

RC1: 14)

p.10 l. 8 What is the experimental evidence that the prism that orders water better is also a better ice nucleator? Were any experiments performed? Couldn't the ageing process and the contact with acid also just roughen the surface and that is the reason why it nucleates better?

*AC1:14)*

*This is a good question and we have clarified this point in the revised version (p. 10, l. 2-4).*

*The experimental evidence that a surface that orders water better "could be" a better ice nucleator has been recently reported (Abdelmonem et al., 2017b; Abdelmonem et al., 2015; Yang et al., 2011). Other parameters, e.g. roughness, porosity, steps…, are not excluded. We believe that all surface properties influence the ice nucleation ability though with different weights. We also believe that the way these properties influence the interaction with water molecules at the surface is the key to the overall effect. In our previous work (Abdelmonem et al., 2017a) we combined freezing assays and SFG characterizations to study the effect of surface charge on the heterogeneous ice-nucleation ability of α-alumina (0001) surfaces. We are not generalizing our former observation on that particular surface (i.e. the α-alumina (0001)), but only recall an existence of evidence.*

*That "Water ordering leads to better ice nucleation condition" is not that straight forward. When a surface is able to create an ordering compatible with the structure of a crystalline phase, it will then promote the nucleation of the corresponding phase particularly if the induced ordering further matches the crystalline structure at a higher degree (Bi et al., 2017). A surface may exhibit the ordering patterns that resemble the structure of ice. Therefore, water layers bound to surfaces may be ice-like, providing a template for ice to nucleate (Bi et al., 2016). Based on these facts we suggest that the re-adsorbed dissolution products have an arrangement on its surface as close as possible to that of water molecules in some low index plane of ice.*

*There is a lack in the literature on what happens at the surface on the molecular level and our approach is applied in ice nucleation studies only since few years ago. With this work we try to attract the attention of theoreticians who can simulate this re-adsorption of dissolution products and their arrangement on the surface. Indeed we have proposed MD simulations with the group of Molecular Modelling and Computer Simulations, Clemson University, as future cooperation.*

RC1: 15)

p.10 Fig.5 Since the observed effect is not very pronounced. It would be good toadd error bars in the Figure or provide information on how reproducible the trend is.

*AC1:15)*

*The authors agree with the Reviewer, this information was missing in the figure caption. Repeating the experiment showed the same trend over the whole range of cycles with an average standard deviation of 1.3. The information has been added in the figure caption with the corresponding error bars on Figure 5 in the revised version of the manuscript.*

RC1: 16)

p.11 l. 15 The results of the following study should be added to the discussion. Rehl et al. 2019 New Insights into χ(3) Measurements: Comparing Nonresonant Second Harmonic Generation and Resonant Sum Frequency Generation at the Silica/Aqueous Electrolyte Interface, JPCA.

*AC1:16)*

*The study of Rehl et al. 2019, although interesting, is not relevant to our study. Rehl et al. compare the SHG and SFG from technical point of view. Neither the sample (IR-grade Silica) nor the solution (0.5 M NaCl) are the same as ours. Even the temperature effect was not discussed. They wanted to find the origin of inconsistencies between SHG and SFG that have arisen when comparing experiments on silica at high electrolyte concentrations. Discussing their results in our manuscript will confuse the reader and deviate from the main study.*

*However, this paper includes one very useful information which is "SHG is more sensitive to the number density of aligned water, particularly at low pH and ionic strength". As already mentioned in the first paragraph in the "Results and Discussion" section, we eliminate the $\chi^{(3)}$ effect by choosing pH = 3, and we also add no salts. We have cited this paper in this context in the revised manuscript (p. 4, l. 26-27).*

RC1: 17)

p.13 l. 33 Is there an explanation why the ice signal shows such strong variations.

*AC1:17)*

*So far we have no explanation for the strong fluctuation in the ice signal after freezing. This will need further investigations on the ice structure after freezing. One expected scenario is the formation of free OH group after freezing (as we observed on Sapphire 110 surface using SFG, not published yet). The role of the formation of this free OH is not yet known. We added the following sentence to p.13. l. 35-36: "This ice signal fluctuation merits further investigations particularly using SFG which gives details on the individual contributions of different interfacial species from their resonant vibrations.". We also merged the "Ice signal" and "Confined liquid signal" sessions.*

RC1: 18)

p. 13. l. 35 Is it possible to estimate the thickness of the liquid film? I would assume this could provide very useful information as one could estimate the pH of this solution which should be much lower due to the freeze concentration and likely dissolves the silica even faster.

*AC1:18)*

*Not with this technique. The maximum penetration depth in our geometry is ~ 400 nm. The layer thickness exceeds this distance very shortly after the freezing point (= transient peak time).*

RC1: 19)

p. 14 l. 28 "this study is expected to benefit"… The connection of the results of the current study and its implications for atmospheric research are not clear.

*AC1:19)*

*The authors have answered on this question in AC1 (second paragraph)*

**II. Point-to-point response to Referee 2 (RC2)**

RC2: 1)

In Fig. 2, 3, and 7, it could be more intuitive to use the temperature as the x axis, instead of the time.

*AC2: 1)*

*The authors would like to thank the Reviewer for his practical suggestion. Indeed plotting the data as a function of temperature will be more perceptive. However, this is only possible in the range of scan where the independent parameter (temperature in this case) is single-valued, otherwise the plot will be confusing, e.g. Fig. AC2.1.*

*Since we focus in this manuscript on the water restructuring upon cooling, plotting the SHG liquid signal as a function of temperature during cooling is the correct choice. Since Figure 3 is a subset of Figure 2, we have only replaced Figure 3 with a new one that now includes temperature as the x axis (Figure AC2.2).*

[Figure]

*Figure AC2.1: This figure shows haw Figure 3 would look like if we plot the SHG signal of the complete scans with temperature is the x-axis.*

[Figure]

*Figure AC2.2: SHG liquid signal as a function of time and temperature during cooling. Figure 3 in the manuscript has been replaced with this Figure.*

*Figure 7 cannot be plotted using temperature as the x axis because the temperature around the freezing and melting peaks is almost constant (i.e. again a not single-valued relation).*

RC2: 2)

The data presented in the manuscript may not directly related to "cloud history", as indicated by the title.

*AC2: 2)*

*As mentioned in the abstract and introduction, an aerosol-containing cloud droplet can go through different freeze-melt or evaporation-condensation cycles, so that not only the aerosol surface structure may change, but also ionic strength and pH of the cloud droplet. We conclude that the cloud history may affect the contained aerosol particles. During cloud formation, water vapor condenses on aerosol particles forming liquid suspended water droplets in about a 100% RH environment. Cloud droplets are constantly forming and dissipating. Depending on the atmospheric conditions (e.g. temperature, RH, or air draft) the cloud droplet change its size. In case of temperature increase, cloud mixing with drier air, or air sinking within the cloud, cloud droplets may evaporate (may also totally dissipate). Under these conditions the acidic, basic components or ionic strength in the cloud droplet will reach extreme values. We show in this manuscript that this may significantly change the surface properties of mineral oxide aerosols. However, we agree with the Reviewer that the title of the original manuscript could be improved. The manuscript title has been changed to "Cloud history can change water-ice-surface interactions of oxide mineral aerosols: a case study on silica".*

**III. Revised manuscript with tracked changes**

[revised manuscript text omitted]

SI3 SHG vs. Cycle number during cooling at different temperatures

Fig 4c in the manuscript looks identical to 4b. It is our intention to show that at the onset point, Fig 4c, there is nothing exceptional happens although the onset temperatures are different.

In Fig. S3 we show the change in signal with aging at other temperatures. Since the paper discusses the restructuring of water upon cooling and relates this to the freezing process, we select here a set of temperatures during cooling to plot the signal as a function of cycle number. Figure S3 shows the averaged SHG liquid signal as a function of TP cycle number at five different temperatures on the cooling path. The minimum points occur at lower cycle numbers for lower temperatures (summarized in table S1). The closest cycle number to the liquid signal minimum decreases with temperature. This assists our conclusion that cooling favors the uptake of dissolved silica (i.e. adsorption).

[Figure]

[Figure]

| Time with respect to scan start (s) | Temperature (°C) | Closest cycle no. to minimum signal |
|---|---|---|
| a) 0 sec | 20 | 7 |
| b) 120 sec | 10 | 6 |
| c) 240 sec | 0 | 5 |
| d) 480 sec | -20 | 4 |
| e) 630 sec | -32 | 3 |

Table S1: The selected temperatures in Figure S3 and the corresponding TP cycle number of minimum SHG liquid signal.

Figure S3: Upper panel: a sample plot of SHG vs. cycle number during cooling. Lower panel: The averaged SHG liquid signal as a function of TP cycle number at five different temperatures during cooling before freezing. The red lines on the plots are guiding lines through the data points.

**S34 The observed fast aging, is it a matter of time only?**

[Figure]

Figure S: SHG signal at pH3 solution-silica interface as a function of TP cycle number (a) and time (b) of liquid signal at 20 °C during repeating the freezing-melting TP. The dashed red lines are trend lines. CS26 and CS27 denote data points that lie on the trend line in (a) but not in (b). This shows that the significant aging we observe in this work arises from the freezing-melting process and not from the time the sample being in contact with solution. Furthermore, as discussed in section S1, the measured silica concentration should favor further dissolution at room temperature, but this does not explain the SHG data.

[Figure]

Figure S45: SHG signal as a function of pH for our fresh sample (before aging) at 20 °C.

[Figure]

Figure S56: SHG signal at pH3 solution-silica interface as a function of time for the C27 run which was carried out at cooling and heating rates (= 1 °C / min) slower than the standard TP.

[Figure]

Figure S67: Photo (top view) of the bulk ice formed after freezing and waiting for 5 min at -40 °C. The freezing starts at the surface of the sample and grows at the expense of water on the other side. The size of the bulk ice is proportional to the time of keeping the system supercooled after the freezing event. The ice piece is stuck in the cell opening and the process of melting and departing from the sample neighborhood depends on the cell geometry and bulk ice size as well as the thermal conditions. All these parameters affect the appearance and disappearance of the confined liquid signal.